# DynFed: Dynamic Test-Time Adaptation for Federated Learning with Adaptive Rate Networks

## Abstract

Test-Time Personalized Federated Learning (TTPFL) has emerged as a promising approach for adapting models to distribution shifts in federated learning (FL) environments without relying on labeled data during testing. However, existing methods often struggle with heterogeneous shifts across clients and lack the flexibility to handle diverse distribution changes effectively. In this paper, we introduce DynFed, a novel algorithm that dynamically optimizes test-time adaptation (TTA) in FL scenarios with heterogeneous distribution shifts. Our method leverages Adaptive Rate Networks (ARNs) (or more generally, adaptive rate functions) to generate client-specific adaptation rates, enabling more effective handling of diverse shift types, including label skew and feature shifts. DynFed employs an innovative iterative adaptation process, where adaptation rates are continuously refined based on the current adaptation state using the ARN function, without direct access to raw client data. Crucially, we uncover a fundamental dichotomy: optimal adaptation strategies for one-type and multi-type distribution shifts can be diametrically opposed. DynFed navigates this challenge by automatically adjusting its approach based on the nature of the encountered shifts. Extensive experiments demonstrate that DynFed significantly outperforms existing TTPFL and TTA methods across various shift scenarios. Our theoretical analysis provides convergence and generalization guarantees for our approach and justifies the need for adaptive mechanisms. Our method shows particularly robust performance in complex multi-type shift environments, where previous approaches often struggle. This work opens new avenues for adaptive and resilient FL in real-world applications where distribution shifts are diverse and unpredictable.

## 1 Introduction

Federated Learning (FL) has emerged as a powerful paradigm for distributed machine learning, enabling models to be trained across multiple decentralized edge devices or servers holding local data samples without the need to exchange them McMahan et al. (2017); Zhao et al. (2018); Mohri et al. (2019); Wang et al. (2021b; 2023; 2024c;b;a). This paradigm ensures data privacy and security by keeping data localized. However, the performance of FL models often degrades when confronted with distribution shifts between training and test data, a challenge exacerbated by the heterogeneous nature of client data in real-world scenarios.

Test-time Adaptation (TTA) has shown promise in addressing this issue by allowing models to adapt to new distributions during inference Zhang et al. (2022); Wang et al. (2021a; 2022). TTA methods enable models to adjust to unseen data distributions, thus enhancing robustness and performance. Recently, Test-time Personalized Federated Learning (TTPFL) has been proposed to combine the benefits of TTA with FL Bao et al. (2024), allowing for unsupervised local adaptation of global models during test time. TTPFL methods provide a framework for individualized model adjustments based on the local data characteristics of each client. However, Existing methods for handling distribution heterogeneity in FL face a critical limitation as depicted as Figure 1 (b). These approaches typically operate under the assumption that all clients encounter data conditions of a consistent type, known as one-type distribution heterogeneity.

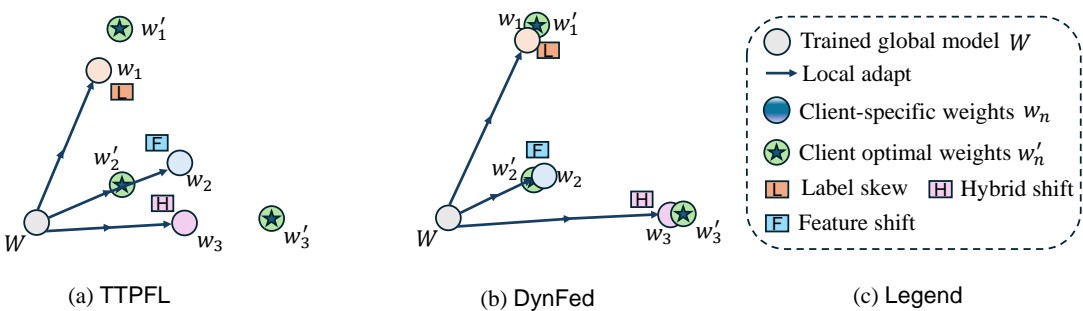

Figure 1: Comparison of TTPFL and DynFed adaptation processes. (a) TTPFL applies uniform adaptation across all clients, resulting in identical local models after each round. (b) DynFed employs client-specific adaptation, leading to diverse local models tailored to each client's distribution. The varying sizes of local model weights ($w_1$, $w_2$, $w_3$) in (b) illustrate the personalized nature of DynFed's adaptation. (c) Legend explaining the symbols used in the figure, where $W$ represents the global model, and $w_n$ denotes $n$-th client-specific weights after adaptation.

This implies that the heterogeneous distributions across clients are uniform, such as all clients experiencing either label skew or all experiencing feature shift. In real-world scenarios, it is impossible to preset the type of heterogeneous distribution each client may encounter. Multi-type distribution heterogeneity is common in FL settings, where some clients may face label skew while others face feature shift. In response to this complex heterogeneous environment, current methods employ fixed adaptation rates for all clients and model components, severely restricting their ability to effectively adapt to heterogeneous distribution shifts. These fixed rates fail to account for the varying degrees of distribution shifts experienced by different clients, resulting in suboptimal adaptation and performance. To address this challenge, we introduce Dyn-Fed, a novel algorithm designed to dynamically optimize test-time adaptation in FL system characterized by heterogeneous distribution shifts. Our approach leverages Adaptive Rate Networks (ARNs) (or functions) to generate client-specific adaptation rates, enabling more effective handling of diverse shift types. The comparison illustrated in Figure 1 highlights the key advantage of our proposed DynFed framework over traditional TTPFL. Our method facilitate client-specific, dynamic adaptation of the global model, in contrast to the uniform adaptation approach employed by conventional TTPFL methods. DynFed employs an innovative iterative adaptation process, where ARNs continuously refine adaptation rates based on the current adaptation state. This process ensures that the adaptation rates are tailored to the specific batch data of each client, thereby enhancing the overall performance and robustness of the federated system in the presence of diverse distribution shifts. The main contributions of our work are as follows:

- We propose a new framework for TTPFL that explicitly accounts for multi-type distribution shifts across clients, including both label skew and feature shifts.

- We introduce Adaptive Rate Networks (ARNs) (or functions) that incrementally generate personalized adaptation rates for each client and model component, significantly improving upon fixed-rate adaptation methods.

- We uncover a critical insight, supported by theoretical arguments (Theorem 3.3), that the optimal adaptation strategies for one-type and multi-type distribution shifts can be fundamentally opposed. Our method, DynFed, effectively navigates this dichotomy by automatically adjusting its strategy based on the nature of the distribution shifts encountered.

- Extensive experiments on various FL benchmarks show that DynFed significantly outperforms traditional TTA and the state-of-the-art TTPFL methods across different types of distribution shifts. Our theoretical analysis provides convergence and generalization guarantees, lending support to the stability and effectiveness of our approach.

In general, our work represents a significant step towards making the TTPFL framework more robust and adaptable in real-world scenarios, where distribution shifts are diverse and unpredictable. By addressing the limitations of current TTPFL methods, DynFed paves the way for more effective and flexible FL systems capable of handling the complexities of real-world data distributions. The rest of this paper is organized as follows: We first review related work in Section 2. Section 3 formally defines the problem setting. We then detail our proposed method in Section 4. Experimental results are presented in Section 5, followed by theoretical analysis in Section 4.4 and the Appendix.

## 2 Related Work

### 2.1 Federated Learning.

Federated Learning (FL) has emerged as a transformative paradigm in distributed machine learning, addressing critical concerns of privacy and data locality. Introduced by McMahan et al. (2017), FL enables model training across decentralized edge devices or servers holding local data samples, without the need for direct data exchange. The seminal FedAvg algorithm, which aggregates locally computed gradients to update a global model, laid the foundation for this field. Since its inception, FL has undergone rapid development across various aspects, including communication efficiency Konečný et al. (2016), privacy preservation Geyer et al. (2017).

The core challenge in real-world FL applications is data heterogeneity, commonly referred to as the non-IID (not independently and identically distributed) problem Zhao et al. (2018). This issue arises as clients often possess diverse label and feature distributions, reflecting their varied behaviors and habits Tan et al. (2022b); Luo et al. (2021). To address this challenge, several innovative approaches have proposed. Clustering-based FL methods aim to group clients based on their data similarity Ghosh et al. (2019); Ma et al. (2022), allowing for more targeted model updates. Concurrently, meta-learning techniques have been leveraged to enhance the personalization capabilities of local models Fallah et al. (2020). Additionally, model decoupling schemes have been introduced to facilitate better personalization while maintaining the benefits of collaborative learning Tan et al. (2023); Bao et al. (2023); Baek et al. (2023).

### 2.2 Test-Time Adaption.

Test-Time Adaptation (TTA) has emerged as a crucial method in machine learning for addressing distribution shifts between training and test data without necessitating model retraining or access to labeled test samples. The core principle of TTA involves adapting the model during inference using only unlabeled test data. Various approaches have been developed to tackle this challenge effectively. Entropy minimization, introduced by Wang et al. (2021a), stands as a fundamental TTA technique. This method fine-tunes the model to generate more confident predictions on test data, thereby aligning the model with the target distribution. Building upon this concept, self-supervised learning has been integrated into TTA frameworks to more efficiently utilize unlabeled test data Chen et al. (2022). This integration allows for the creation of auxiliary tasks that guide adaptation without relying on explicit labels. Recent advancements in TTA include more sophisticated techniques. SAR Niu et al. (2023) enhances adaptation by eliminating high-gradient samples and promoting weights that lead to flat minima, thus improving generalization. In the realm of pseudo labeling-based methods, PL Lee et al. (2013) refines model parameters using confidently predicted pseudo labels. SHOT Liang et al. (2020) takes a hybrid approach, combining entropy minimization strategies with pseudo labeling techniques to achieve robust adaptation. These diverse methods collectively represent the ongoing efforts to develop TTA approaches that are both effective and widely applicable across various domains and types of distribution shifts.

### 2.3 Test-Time Adaptation in Federated Learning.

Test-Time Adaptation in Federated Learning ( TTA-FL) presents a more complex scenario compared to traditional end-to-end TTA, primarily due to the inherent data distribution challenges in FL environments Jiang & Lin (2023); Tan et al. (2024); Wan et al. (2024). While conventional TTA methods focus on adapting

models to shifts between a single source and target domain, TTA-FL must contend with multiple, heterogeneous client distributions, It can also be called Test-Time Personlized Federated Learning ( TTPFL) Bao et al. (2024), this means that the source model is a trained global model, but personalized to the data of individual clients. Existing methods assume that all clients face consistent data type conditions, referred to as one-type distribution heterogeneity, where the heterogeneous distributions on each client are the same, such as either uniformly label skew or uniformly feature shift. However, in real-world scenarios, it is impossible to preset the type of heterogeneous distribution on each client. Multi-type distribution heterogeneity is quite common, where in a FL scenario, some clients experience label skew while others experience feature shift. Existing TTPFL Bao et al. (2024) methods utilize a uniform adaptation rate for all participating clients, which is evidently inadequate for addressing the needs of multi-type distribution heterogeneity scenarios.

## 3 Preliminaries

In this section, we first introduced the notion of TTPFL antecedents. Then the limitations of the current framework for TTPFL are discussed.

### 3.1 Test-time personalized federated learning

#### 3.1.1 Global and Personalized Federated Learning

Global Federated Learning (GFL) aims to find a single global model that minimizes the expected loss over the client population:

$$\mathcal{L}(\mathbf{w}_G) = \mathbb{E}_{P \sim Q}[\mathcal{L}_P(\mathbf{w}_G)], \quad \text{where} \quad \mathcal{L}_P(\mathbf{w}_G) = \mathbb{E}_{(x,y) \in P} \ell(f(x; \mathbf{w}_G); y) \tag{1}$$

where $\mathbf{w}_G$ represents the parameters of the global model, $\mathcal{L}(\mathbf{w}_G)$ is the overall loss function for the global model, $P$ represents the data distribution of an individual client, $Q$ is the distribution of client distributions, $\mathcal{L}_P(\mathbf{w}_G)$ is the loss function for a specific client with distribution $P$, $x$ and $y$ are input features and labels respectively, $\ell(\cdot; \cdot)$ represents the loss function (e.g., cross-entropy), $f(\cdot; \cdot)$ represents the model function parameterized by $\mathbf{w}_G$, and $\mathbb{E}[\cdot]$ denotes the expectation operator. GFL mandates that each client uses the same global model for prediction, precluding adaptation to each client's unique data distribution. In contrast, personalized federated learning (PFL) customizes the global model $\mathbf{w}_G$ using the client's labeled data and employs the personalized model for prediction. However, most PFL algorithms Tan et al. (2022a); Deng et al. (2020); Fallah et al. (2020) assume that the target client also possesses additional labeled data, a stronger assumption compared to GFL.

#### 3.1.2 Test-Time Personalized Federated Learning

In the paper Bao et al. (2024), they introduced a novel paradigm called test-time personalization federated learning (aka TTPFL). TTPFL focuses on adapting a trained global model to each target client's unlabeled data during test-time, utilizing an adaptation rule A that operates solely on unlabeled data. The objective function can be formulated as:

$$\mathcal{L}(\mathbf{w}_G, A) = \mathbb{E}_{P \sim Q}[\mathcal{L}_P(\mathbf{w}_G, A)], \quad \text{where} \quad \mathcal{L}_P(\mathbf{w}_G, A) = \mathbb{E}_{(x,y) \in P} \ell(f(x; A(\mathbf{w}_G, X)); y) \tag{2}$$

which can be unbiasedly estimated by the average loss over $M$ target clients unseen during training:

$$\hat{\mathcal{L}}(\mathbf{w}_G, A) = \frac{1}{M} \sum_{j=1}^{M} \hat{\mathcal{L}}_{T_j}(\mathbf{w}_G, A), \quad \text{where} \quad \hat{\mathcal{L}}_{T_j}(\mathbf{w}_G, A) = \frac{1}{m_j} \sum_{r=1}^{m_j} \ell(f(x_{T_j}^r; A(\mathbf{w}_G, X_{T_j}^r)); y_{T_j}^r) \tag{3}$$

where $\hat{\mathcal{L}}_{P_j}(\mathbf{w}_G, A)$ denotes the empirical loss on the $j$-th target client. The adaptation rule A modifies the global model using unlabeled samples $X_{T_j}^r$. TTPFL consider two standard settings: Test-Time Batch Adaptation (TTBA) and Online Test-Time Adaptation (OTTA)Liang et al. (2024). TTBA individually adapts the global model to each batch of unlabeled samples, where $X_{T_j}^r$ represents the data batch containing $x_{T_j}^r$. OTTA adapts the global model in an online manner, where $X_{T_j}^r$ encompasses all data batches arriving before or concurrently with $x_{T_j}^r$.

### 3.2 Limitation of TTPFL

Despite the promising advancements in TTPFL, current approaches suffer from several key limitations that hinder their effectiveness in real-world scenarios. Existing methods often presume a uniform distribution shift across all clients, neglecting the reality of heterogeneous shifts in federated environments. This over-simplification is compounded by the prevalent use of static, predefined adaptation rates for all clients and model components, significantly limiting the ability to adapt to diverse and dynamic distribution shifts. To formally express this limitation, we can formulate the TTPFL objective as:

$$\hat{\mathcal{L}}(\mathbf{w}_G, \{\boldsymbol{\alpha}_j\}_{j=1}^M, \{\delta_j\}_{j=1}^M) = \frac{1}{M} \sum_{j=1}^M \hat{\mathcal{L}}_{P_j}(\mathbf{w}_G, \boldsymbol{\alpha}_j, \delta_j) \tag{4}$$

where $\mathbf{w}_G$ represents the global model parameters, $\{\boldsymbol{\alpha}_j\}_{j=1}^M$ denotes the set of adaptation rate vectors for $M$ clients, $\{\delta_j\}_{j=1}^M$ represents the set of distribution shifts for each client, and $\hat{L}_{P_j}(\mathbf{w}_G, \boldsymbol{\alpha}_j, \delta_j)$ is the empirical loss on client $j$ using adaptation rate vector $\boldsymbol{\alpha}_j$ under distribution shift $\delta_j$. This formulation highlights the key limitation: each client $j$ may require a different adaptation rate vector $\boldsymbol{\alpha}_j$ to handle its specific distribution shift $\delta_j$, which is challenging to achieve effectively in practice, especially for multi-type distribution shifts and dynamically changing environments. Furthermore, many TTPFL algorithms lack mechanisms to gradually adjust adaptation strategies based on evolving client data distributions, potentially leading to suboptimal performance over time Zhang et al.. The inefficient use of historical adaptation information also represents a missed opportunity for more informed and efficient adaptation processes. These interrelated issues collectively impair the effectiveness of TTPFL in complex, real-world federated learning environments Zeng et al. (2023); Zhang et al. (2024), where distribution shifts are often multifaceted, diverse, and continually evolving.

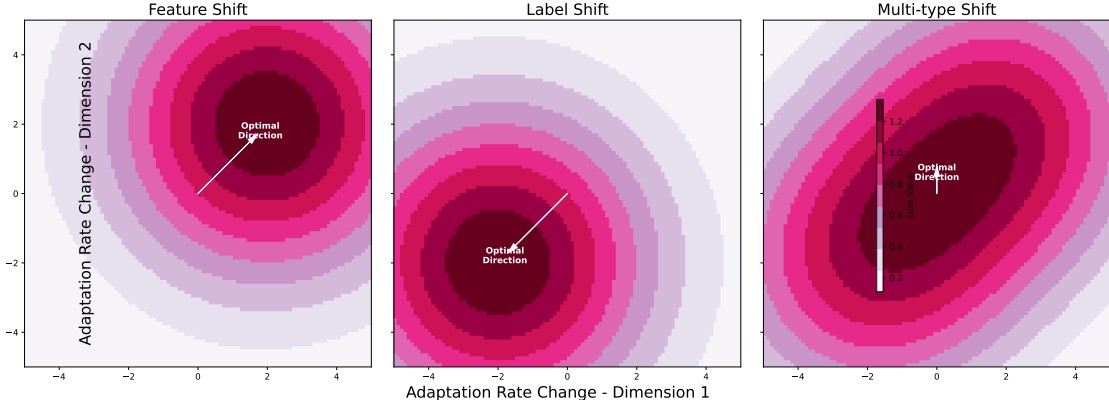

Figure 2: Loss landscapes and optimal adaptation directions for different types of distribution shifts. Left: Feature shift only. Middle: Label shift only. Right: Multi-type distribution shift (combined feature and label shifts in different clients, And the two-type shifts don't overlap). White arrows indicate the optimal adaptation directions. Note that the optimal direction for multi-type shift differs significantly from single-type shifts in FL, justifying the need for different adaptive scaling in participant clients.

### 3.3 Challenges

The primary challenge in TTPFL lies in effectively addressing the not only one-type distribution hetero-geneity prevalent in real-world FL environments. This challenge is significantly more complex than dealing with one-type distribution heterogeneity, as the optimization directions for different types of shifts can be inconsistent or even conflicting. Our experiment observations, as illustrated in Figure 2, reveal a critical insight: Even the direction of the adaption is not consistent across one-type distribution shifts. When multiple distinct distribution shifts occur simultaneously across different clients, the optimal adaptation direction

without server communication can be diametrically opposed to the direction in scenarios with a single, uniform shift. This phenomenon underscores the inadequacy of conventional TTPFL methods that assume a homogeneous distribution shift across all clients.

[Adaptation Strategy Dichotomy] Let $\mathcal{S}_1, \ldots, \mathcal{S}_N$ denote a set of clients experiencing a one-type distribution shift (either purely feature or purely label shift), and let $\mathcal{M}_1, \ldots, \mathcal{M}_N$ denote a set of clients experiencing multi-type (mixed) distribution shifts. Then, under strong convexity assumptions (detailed in Appendix A), the optimal adaptation rate vectors $\boldsymbol{\alpha}_\mathcal{S}^*$ for one-type shifts and $\boldsymbol{\alpha}_\mathcal{M}^*$ for multi-type shifts satisfy

$$\boldsymbol{\alpha}_\mathcal{S}^* \cdot \boldsymbol{\alpha}_\mathcal{M}^* < 0, \tag{5}$$

where "$\cdot$" denotes the dot product, indicating that the optimal adaptation directions are negatively correlated.

This theorem mathematically confirms our experimental observations depicted in Figure 2: the optimal adaptation strategies for different types of distribution shifts can be diametrically opposed. This poses a fundamental challenge for federated learning systems that must simultaneously handle both one-type and multi-type distribution shifts across clients.

The core challenge, therefore, is to develop a unified approach that can effectively handle diverse types of distribution shifts occurring across different clients. This approach must be capable of gradually adjusting adaptation strategies to accommodate these heterogeneous shifts. Furthermore, this solution must operate within the constraints of FL, maintaining data privacy and minimizing communication overhead. The ability to address these multifaceted challenges without compromising the core principles of FL is the cornerstone of our proposed DynFed method.

## 4 Method

In this section, we introduce DynFed, a novel framework that gradually generates client-specific adaptation rates for each module in FL settings. Our approach addresses the challenges of heterogeneous distribution shifts across clients, offering a more flexible and efficient solution compared to existing methods. We detail the training and testing phases of DynFed, followed by a discussion of its advantages and theoretical properties.

### 4.1 Training Phase: Learning Adaptation Rates with Source Clients

Following the training stage of Bao et al. (2024), our training phase builds upon the TTPFL framework while introducing key innovations. DynFed employs a communication protocol similar to FedAvg McMahan et al. (2017) to initialize and refine adaptation rates. The training process is iterative, with each round consisting of local unsupervised adaptation, supervised refinement, and server aggregation. Similar to previous works, we consider the model processes a data batch $X_{Si}^k = \{x_{Si,k,b}\}_{b=1}^B$ at a time where $B$ is the batch size, $i$ is the client index and $k$ is the batch index. In the following, we omit the superscript $Si$ for clarity, e.g. $X_{Si}^k \rightarrow X^k$, as unsupervised adaptation and supervised refinement operate identically across all source clients.

*Unsupervised Adaptation:* We consider a neural network model $f(\cdot; \mathbf{w}_G)$ with global model parameter $\mathbf{w}_G \in \mathbb{R}^D$. The network comprises $d$ modules, each with parameters $w^{[1]}, \ldots, w^{[d]}$. During unsupervised adaptation, we allow each module $w^{[l]}$ to have a distinct adaptation rate $\alpha^{[l]}$, enabling fine-grained control over the adaptation process. Here, $\boldsymbol{\alpha} \in \mathbb{R}^d$ denotes the vector of adaptation rates for all modules for the current client and batch.

*Update Trainable Parameters:* For each trainable parameter module $w^{[l]}$, we compute the unsupervised update direction as the negative gradient of the entropy loss:

$$h_k^{[l]} = -\nabla_{w^{[l]}} \ell_H(f(X_k; \mathbf{w}_G)) \tag{6}$$

where $\ell_H(\hat{Y}) = -\frac{1}{B} \sum_{b=1}^B \sum_{c=1}^C \hat{y}_{b,c} \log \hat{y}_{b,c}$ is the entropy loss. This approach allows the model to adapt without requiring labeled data, making it suitable for real-world scenarios where labels may be scarce or unavailable.

*Update Running Statistics:* For batch normalization layers, which play a crucial role in adapting to distribution shifts (see Proposition 4.4.3), we define the update direction as the difference between the current batch statistics and the global running statistics:

$$h_k^{[l]} = \hat{w}_k^{[l]} - w_G^{[l]} \tag{7}$$

where $\hat{w}_k^{[l]}$ represents the batch normalization statistics (mean and variance) estimated from the current batch of inputs $X_k$, and $w_G^{[l]}$ represents the global running statistics maintained by the BN layer. This update ensures that the batch normalization layers can effectively adapt to the local data distribution of each client.

After computing the update directions $\mathbf{h}_k = \{h_k^{[l]}\}_{l=1}^d$, each module is updated using its corresponding adaptation rate $\alpha^{[l]}$ from the vector $\boldsymbol{\alpha}$. Let $A(\boldsymbol{\alpha})$ represent the operation of applying these module-specific rates to the corresponding update directions, typically via element-wise multiplication if gradients are structured by module, i.e., $(A(\boldsymbol{\alpha}) \odot \mathbf{h}_k)^{[l]} = \alpha^{[l]} h_k^{[l]}$. The update is:

$$w_k^{[l]} \leftarrow w_G^{[l]} + \alpha^{[l]} h_k^{[l]} \quad (\text{or } \mathbf{w}_k \leftarrow \mathbf{w}_G + A(\boldsymbol{\alpha}) \odot \mathbf{h}_k) \tag{8}$$

Note that $\alpha^{[l]}$ represents the adaptation rate for module $l$ for the current client and batch.

*Supervised Refinement:* Following the unsupervised adaptation, we refine the adaptation rates $\boldsymbol{\alpha}$ using gradient descent on the cross-entropy loss computed with the adapted parameters $\mathbf{w}_k$ and the corresponding labels $Y_k$:

$$\boldsymbol{\alpha} \leftarrow \boldsymbol{\alpha} - \eta_\alpha \nabla_{\boldsymbol{\alpha}} \ell_{CE}(f(X_k; \mathbf{w}_k), Y_k) \tag{9}$$

where $\eta_\alpha$ is the learning rate for adaptation rates, and $\ell_{CE}$ is the cross-entropy loss. This step refines the adaptation rates $\boldsymbol{\alpha}$ by minimizing the supervised loss on the post-adaptation parameters $\mathbf{w}_k$. The gradient $\nabla_{\boldsymbol{\alpha}} \ell_{CE}$ is computed via the chain rule, backpropagating through the adaptation step (Eq. 8) to update $\boldsymbol{\alpha}$ such that future unsupervised adaptations using these rates lead to better supervised performance. This mechanism allows the model to learn how to adapt effectively based on labeled source data.

*Server Aggregation:* To leverage the collective knowledge of all clients and improve generalization, we employ federated aggregation (e.g., FedAvg) to periodically combine the locally refined adaptation rates $\boldsymbol{\alpha}_i$ from different source clients $i$. This step is crucial for ensuring that the learned adaptation strategies are effective across a diverse range of clients and data distributions.

## 4.2   Testing Phase: Exploiting Dynamic Adaptation on Target Clients

During the testing phase, each target client receives the global model $\mathbf{w}_G$ and the aggregated initial adaptation rates $\boldsymbol{\alpha}_G$ learned during training. The client also uses a predefined Adaptive Rate Network (ARN) function, denoted as $g(\cdot)$, which dictates how adaptation rates evolve based on their current state. We consider two settings, following TTPFL Bao et al. (2024): DynFed-batch for test-time batch adaptation (TTBA) and DynFed-online for online test-time adaptation (OTTA).

*Adaptive Rate Network (ARN) Function:* The cornerstone of our dynamic adaptation mechanism during testing is the ARN function $g(\cdot)$. This function takes the current adaptation rates $\boldsymbol{\alpha}_t$ as input and outputs a target adaptation rate vector. In our implementation, we use a predefined, fixed structure for $g$, such as a small Multi-Layer Perceptron (MLP) or even simpler functions (e.g., identity mapping, $g(\boldsymbol{\alpha}_t) = \boldsymbol{\alpha}_t$, relying solely on momentum and the learned initial $\boldsymbol{\alpha}_G$). The parameters $\theta$ of $g$, if any, are not trained via Eq. 9; instead, the adaptation strategy is learned entirely through the refinement and aggregation of the rates $\boldsymbol{\alpha}$ themselves during the training phase. The function $g$ defines the dynamics of how rates evolve from their current state.

We formally define the ARN function $g$ used in our experiments as a compact neural network with fixed parameters $\theta$ (or no parameters if $g$ is identity):

$$g(\boldsymbol{\alpha}_t) = \tau \cdot \sigma_{\text{sigmoid}}(f_L(\ldots f_2(f_1(\boldsymbol{\alpha}_t))\ldots;\theta)) \tag{10}$$

where $f_i$ represents the $i$-th layer of the fixed network structure, $\sigma_{\text{sigmoid}}$ is the sigmoid activation function applied at the output, and $\tau$ is a scaling factor hyperparameter that ensures the output adaptation rates fall within an appropriate range. Each hidden layer $f_i$ (for $i < L$) is defined as:

$$f_i(z) = \sigma_{\text{ReLU}}(W_i z + b_i) \tag{11}$$

Here, $W_i$ and $b_i$ are components of the fixed parameters $\theta$. This structure allows the rate dynamics to follow a non-linear trajectory influenced by the current rate state $\boldsymbol{\alpha}_t$.

*Dynamic Adaptation Rate Computation:* We introduce a novel mechanism to dynamically compute the adaptation rates during testing. The process starts with the initial rates $\boldsymbol{\alpha}_G$ learned during training, which are then iteratively updated for each batch of test data on each client. Let $\boldsymbol{\alpha}_t$ denote the adaptation rate vector at time step $t$. We compute the rates for the next time step $t+1$ using the predefined ARN function $g$:

$$\boldsymbol{\alpha}_{t+1} = \lambda \cdot \boldsymbol{\alpha}_t + (1 - \lambda) \cdot g(\boldsymbol{\alpha}_t) \tag{12}$$

where $\lambda \in [0, 1]$ is a hyperparameter controlling the momentum or smoothing between the previous rates and the target rate generated by the ARN function $g$. The initial adaptation rates $\boldsymbol{\alpha}_0$ are set to the aggregated rates $\boldsymbol{\alpha}_G$ from the training phase. This formulation allows for a dynamic evolution of adaptation rates based on their current state via the fixed function $g$ and the learned initial state $\boldsymbol{\alpha}_G$, enabling the model to continuously refine its adaptation approach as it processes more test data.

*DynFed-batch:* In the TTBA setting, each target client processes data batches independently. For each batch $k$ with data $X_{jk}^T$, DynFed-batch first computes the current adaptation rates $\boldsymbol{\alpha}_{\text{current}}$ using Eq. (11) based on the previous rates $\boldsymbol{\alpha}_t$. Then, it computes the unsupervised update direction $h_{jk}^T$ (Eq. 6 or 7) using the global model $\mathbf{w}_G$. Finally, it adapts the model using $w_{jk}^T \leftarrow w_G + A(\boldsymbol{\alpha}_{\text{current}}) \odot h_{jk}^T$ and makes predictions using $w_{jk}^T$. The rates $\boldsymbol{\alpha}_t$ are then updated to $\boldsymbol{\alpha}_{\text{current}}$ for the next batch. This approach allows for quick adaptation to batch-specific characteristics.

*DynFed-online:* For OTTA, data arrives sequentially. To mitigate batch dependency and create more stable adaptation, we adapt based on an average of past update directions:

$$w_k^T \leftarrow w_G + A(\boldsymbol{\alpha}_{\text{current}}) \odot \bar{h}_k^T \quad \text{where} \quad \bar{h}_k^T = \frac{1}{k} \sum_{s=1}^{k} h_s^T \tag{13}$$

Here, $\boldsymbol{\alpha}_{\text{current}}$ is computed using Eq. (11) as in DynFed-batch, $h_s^T$ is the update direction computed for batch $s$ using $\mathbf{w}_G$, and $\bar{h}_k^T$ is the running average of update directions up to batch $k$. This averaging effectively simulates updating with a larger batch size, utilizing historical information while adapting dynamically via $\boldsymbol{\alpha}_{\text{current}}$. This approach allows for more stable adaptation in online scenarios.

Algorithm 1 outlines the training phase, focusing on learning the adaptation rates $\boldsymbol{\alpha}$ on source clients and aggregating them into $\boldsymbol{\alpha}_G$. Algorithm 2 details the testing phase, where the learned initial rates $\boldsymbol{\alpha}_G$ and a predefined ARN function $g(\cdot)$ are used with the momentum update (Eq. 11) to dynamically generate rates $\boldsymbol{\alpha}_{\text{current}}$ for adapting the global model $\mathbf{w}_G$ to target client data.

### 4.3 Discussion

DynFed offers several key advantages over existing TTPFL methods. Firstly, our approach is inherently more flexible due to the use of the dynamic rate update mechanism, which can adjust adaptation rates based on the specific characteristics of each client's data distribution (as represented by the evolving adaptation state $\boldsymbol{\alpha}_t$). This flexibility allows DynFed to handle a wider range of distribution shifts more effectively than static adaptation methods. Secondly, the theoretical underpinning (Theorem 3.3, Propositions 4.4.3, 4.4.3) suggests

---

**Algorithm 1** DynFed-Training

---

**Input**: Initial global model $\mathbf{w}_G$, initial adaptation rates $\boldsymbol{\alpha}_0^G$
**Parameter**: Number of communication rounds $T$, number of clients per round $C$, local epochs $E$, rate learning rate $\eta_\alpha$
**Output**: Final aggregated adaptation rates $\boldsymbol{\alpha}_G^T$

1: Broadcast $\mathbf{w}_G$ and $\boldsymbol{\alpha}_{t-1}^G$ to all source clients (for $t = 1$, use $\boldsymbol{\alpha}_0^G$)
2: **for** communication round $t = 1$ to $T$ **do**
3:     $S_t \leftarrow$ (random set of $C$ source clients)
4:     **for all** source client $S_i \in S_t$ in parallel **do**
5:         $\boldsymbol{\alpha}_i^t \leftarrow \text{CLIENTTRAIN}(S_i, \mathbf{w}_G, \boldsymbol{\alpha}_{t-1}^G)$
6:     **end for**
7:     $\boldsymbol{\alpha}_G^t = \frac{1}{C} \sum_{S_i \in S_t} \boldsymbol{\alpha}_i^t$ {Aggregate adaptation rates}
8: **end for**
9: **return** $\boldsymbol{\alpha}_G^T$

    **ClientTrain**$(S_i, \mathbf{w}_G, \boldsymbol{\alpha}_{\text{init}})$: {Train on client $S_i$}
10: Initialize local rates $\boldsymbol{\alpha} \leftarrow \boldsymbol{\alpha}_{\text{init}}$
11: **for** local epoch $e = 1$ to $E$ **do**
12:     **for all** batch $(X_k^{S_i}, Y_k^{S_i})$ in $D^{S_i}$ **do**
13:         Compute unsupervised update directions $\mathbf{h}_k^{S_i}$ using $\mathbf{w}_G$ (Eq. 6 or 7)
14:         Adapt model: $\mathbf{w}_k^{S_i} \leftarrow \mathbf{w}_G + A(\boldsymbol{\alpha}) \odot \mathbf{h}_k^{S_i}$ (Eq. 8)
15:         Refine rates: $\boldsymbol{\alpha} \leftarrow \boldsymbol{\alpha} - \eta_\alpha \nabla_{\boldsymbol{\alpha}} \ell_{CE}(f(X_k^{S_i}; \mathbf{w}_k^{S_i}), Y_k^{S_i})$ (Eq. 9)
16:     **end for**
17: **end for**
18: **return** $\boldsymbol{\alpha}$ {Return the refined adaptation rates}

---

---

**Algorithm 2** DynFed-Testing

---

**Input**: Target client $T_j$, global model $\mathbf{w}_G$, learned initial adaptation rates $\boldsymbol{\alpha}_G$, predefined ARN function $g(\cdot)$, momentum $\lambda$
**Parameter**: Adaptation mode (TTBA or OTTA)
**Output**: Adapted predictions $\{\hat{Y}_{jk}^T\}$

1: Initialize $\mathbf{h}_{\text{history}} \leftarrow \mathbf{0}$ {For OTTA}
2: Initialize current rates $\boldsymbol{\alpha}_t \leftarrow \boldsymbol{\alpha}_G$
3: **for all** batch $k = 1, 2, \dots$ in test dataset $D_j^T$ **do**
4:     Compute next rates: $\boldsymbol{\alpha}_{\text{current}} = \lambda \cdot \boldsymbol{\alpha}_t + (1 - \lambda) \cdot g(\boldsymbol{\alpha}_t)$ (Eq. 11)
5:     Compute unsupervised update direction $\mathbf{h}_{jk}^T$ using $\mathbf{w}_G$ (Eq. 6 or 7) on batch $X_{jk}^T$
6:     **if** TTBA **then**
7:         Adapt model: $\mathbf{w}_{jk}^T \leftarrow \mathbf{w}_G + A(\boldsymbol{\alpha}_{\text{current}}) \odot \mathbf{h}_{jk}^T$
8:     **else**
9:         Update history: $\mathbf{h}_{\text{history}} \leftarrow \frac{k-1}{k} \mathbf{h}_{\text{history}} + \frac{1}{k} \mathbf{h}_{jk}^T$
10:        Adapt model: $\mathbf{w}_{jk}^T \leftarrow \mathbf{w}_G + A(\boldsymbol{\alpha}_{\text{current}}) \odot \mathbf{h}_{\text{history}}$ (Eq. 12)
11:     **end if**
12:     Make prediction: $\hat{Y}_{jk}^T = f(X_{jk}^T; \mathbf{w}_{jk}^T)$
13:     Update rates for next step: $\boldsymbol{\alpha}_t \leftarrow \boldsymbol{\alpha}_{\text{current}}$
14: **end for**
15: **return** $\{\hat{Y}_{jk}^T\}$

---

that such adaptability is crucial for navigating the conflicting requirements of different shift types. Finally, our experiments (Section 5) demonstrate significant empirical gains, particularly in challenging multi-type shift scenarios.

### 4.4 Theoretical Properties of DynFed

Our approach not only demonstrates empirical improvements but also possesses theoretical justifications. We provide the key theoretical results here, with full proofs deferred to Appendix A. These results establish the stability and generalization capabilities of DynFed and provide insights into the necessity of adaptive mechanisms for heterogeneous shifts.

#### 4.4.1 Convergence Analysis

We analyze the convergence of the training process (Algorithm 1) where adaptation rates are learned. We consider the overall objective across source clients.

[Smoothness] The loss function $\mathcal{L}_{P_j}(\mathbf{w}_G, \boldsymbol{\alpha}_j)$ is $\beta$-smooth with respect to both $\mathbf{w}_G$ and $\boldsymbol{\alpha}_j$.

[Bounded Gradients] The expected squared norms of the stochastic gradients used in Algorithm 1 (averaged over clients and local steps) are bounded with respect to the true gradients of the global loss $\mathcal{L}(\mathbf{w}_G, \{\boldsymbol{\alpha}_j\}_{j=1}^M) = \frac{1}{M}\sum \mathcal{L}_{P_j}(\mathbf{w}_G, \boldsymbol{\alpha}_j)$:

$$\mathbb{E}\left\|\hat{\nabla}_{\mathbf{w}_G} - \nabla_{\mathbf{w}_G}\mathcal{L}(\mathbf{w}_G, \{\boldsymbol{\alpha}_j\})\right\|^2 \le \sigma_w^2,$$

and for each client $j$'s rate vector $\boldsymbol{\alpha}_j$:

$$\mathbb{E}\left\|\hat{\nabla}_{\boldsymbol{\alpha}_j} - \nabla_{\boldsymbol{\alpha}_j}\mathcal{L}(\mathbf{w}_G, \{\boldsymbol{\alpha}_j\})\right\|^2 \le \sigma_\alpha^2.$$

(Note: Here, $\boldsymbol{\alpha}_j$ refers to the adaptation rate vector specific to client $j$ in the context of the overall learning objective, distinct from the time-indexed $\boldsymbol{\alpha}_t$ used within algorithms). [Convexity] The global loss function $\mathcal{L}(\mathbf{w}_G, \{\boldsymbol{\alpha}_j\}_{j=1}^M)$ is convex with respect to the global model parameters $\mathbf{w}_G$ and each client's adaptation rate vector $\boldsymbol{\alpha}_j$.

*Remark on Convexity:* We note that Assumption 3 (Convexity) is standard in the convergence analysis of many optimization algorithms for tractability, although the true loss landscape for deep neural networks is generally non-convex. Our analysis under convexity provides insights into the algorithm's stability and behavior in simpler settings Zhao et al. (2018). Extending the analysis to non-convex settings, potentially using the Polyak-Lojasiewicz (PL) condition, remains an avenue for future work.

[Convergence of DynFed Training] Under Assumptions 1, 2, and 3, with appropriately chosen learning rates for the model parameters ($\eta_w$, implicitly handled by the base FL algorithm) and adaptation rates ($\eta_\alpha \le 1/(2\beta)$ in Algorithm 1), the DynFed training process converges. Specifically, the average squared norm of the gradients of the global loss $\mathcal{L}$ diminishes over $T$ communication rounds, ensuring the algorithm approaches a stationary point. The expected squared gradient norm averaged over $T$ rounds is bounded:

$$\frac{1}{T}\sum_{t=0}^{T-1}\mathbb{E}\left[\|\nabla_{\mathbf{w}_G}\mathcal{L}(\mathbf{w}_G^t, \{\boldsymbol{\alpha}_j^t\}_j)\|^2 + \sum_{j=1}^M \|\nabla_{\boldsymbol{\alpha}_j}\mathcal{L}(\mathbf{w}_G^t, \{\boldsymbol{\alpha}_j^t\}_j)\|^2\right]$$
$$\le \mathcal{O}\left(\frac{\mathbb{E}[V_0]}{T} + \eta_w^2\sigma_w^2 + \eta_\alpha^2\sigma_\alpha^2\right), \tag{14}$$

where $V_0$ is the initial Lyapunov distance to the optimum and the expectation is over the stochasticity of client sampling and local updates. (See Appendix A for the full statement and proof).

This theorem establishes the convergence properties of the DynFed training phase. It guarantees that the algorithm converges to a neighborhood of an optimal solution (for the adaptation rates and potentially the global model, depending on the base FL method), where the size of the neighborhood is determined by the learning rates and the gradient variances. Higher heterogeneity across clients might increase $\sigma_w^2$ and $\sigma_\alpha^2$, potentially enlarging the convergence neighborhood, but convergence is still guaranteed. This result confirms the stability of DynFed's training process, ensuring that the expected gradients remain bounded despite the complexity of learning client-specific adaptation rates via stochastic updates.

#### 4.4.2 Generalization Guarantees

[Generalization of Learned Adaptation Strategy] Let $\mathcal{H} = \{\boldsymbol{\alpha} : \|\boldsymbol{\alpha}\|_2 \leq R\}$ be the hypothesis space of adaptation rate vectors learned during training, $N$ be the number of source clients, and $K$ be the number of data batches per source client used for refinement (Eq. 9). Assuming (1) the model $f$ is $L$-Lipschitz w.r.t. its parameters and (2) the update directions $h^{[l]}$ for each module are bounded by $H$, then for any fixed global model $\mathbf{w}_G$ and any $\epsilon > 0$, the generalization gap between the expected post-adaptation error rate $\varepsilon(\boldsymbol{\alpha})$ over the client population and the empirical error rate $\hat{\varepsilon}(\boldsymbol{\alpha})$ on source clients is bounded with high probability:

$$\Pr\Big(\sup_{\boldsymbol{\alpha} \in \mathcal{H}} |\varepsilon(\boldsymbol{\alpha}) - \hat{\varepsilon}(\boldsymbol{\alpha})| \geq \epsilon\Big) \leq \mathcal{O}\left(\left(\frac{LHRd}{\epsilon}\right)^d \cdot \exp\Big(-\frac{NK\epsilon^2}{poly(d, R, L, H)}\Big)\right), \tag{15}$$

where $d$ is the dimensionality of the adaptation parameter vector $\boldsymbol{\alpha}$. (See Appendix A for details).

This theorem demonstrates that the adaptation strategy learned by DynFed during training can generalize well to unseen target clients, despite its enhanced expressiveness through client-specific adaptation rates. The generalization error depends on the capacity of the adaptation parameter space (controlled by $d$ and $R$) and decreases exponentially with the number of source clients $N$ and the amount of data $K$ used for learning the rates. This ensures that the learned ARN and initial rates $\boldsymbol{\alpha}_G$ are likely to be effective on new clients.

#### 4.4.3 Adaptation Mechanisms for Different Distribution Shifts

The following propositions provide theoretical justification for why different model components require distinct adaptation strategies for different shift types, motivating the module-specific rates $\alpha^{[l]}$ used in DynFed.

[Adapting the Last Layer for Label Shift] Consider a source distribution $p$ and a target distribution $q$ experiencing only label shift, i.e., $p(\mathbf{x}|\mathbf{y}) = q(\mathbf{x}|\mathbf{y})$ and $p(\mathbf{y}) \neq q(\mathbf{y})$. If a neural network classifier $f(\mathbf{x}; \mathbf{w})$ is calibrated on $p$ (outputs match true probabilities $p(\mathbf{y}|\mathbf{x})$), it can be recalibrated for $q$ by adjusting only the bias term of the final layer, specifically by adding $\log(q(\mathbf{y})/p(\mathbf{y}))$. (See Appendix A for proof).

[Adapting BN Layers for Feature Shift] Assume a feature shift causes changes primarily in the first and second-order moments (mean and variance) of feature activations $\mathbf{z}$ at some layer, while higher-order statistics remain relatively stable. This shift can often be effectively mitigated by adapting the running mean and variance statistics of Batch Normalization (BN) layers applied to $\mathbf{z}$. (See Appendix A for proof sketch).

These propositions formally justify why adapting only specific parts of the network (e.g., the last layer for label shift, BN layers for feature shift) can be optimal. DynFed's use of module-specific adaptation rates $\alpha^{[l]}$ allows it to learn and apply these targeted strategies, potentially assigning larger rates to BN layers under feature shift and to the final layer under label shift. The ARN provides the mechanism to dynamically adjust these rates based on the encountered data. Theorem 3.3 further highlights that the *combination* of shifts in a multi-type scenario necessitates a complex, potentially non-intuitive balancing act between these different adaptation needs, reinforcing the requirement for a flexible mechanism like ARN.

## 5 Experiments

### 5.1 Experiments setting

We evaluate our proposed framework on a variety of models, datasets and distribution shifts. We first evaluate on CIFAR-10(-C) with a standard three-way split Yuan et al. (2021): we randomly split the dataset to 300 clients: 240 source clients and 60 target clients. Each source client has 160 training samples and 40 validation samples, while each target client has 200 unlabeled testing samples. We simulate four kinds of distribution shifts: feature shift, label shift, hybrid shift (overlapping feature and label shift on all clients), and multi-type distribution shift (MT shift, where different clients experience different, non-overlapping shifts, e.g., some feature, some label). For feature shift, we follow Hendrycks & Dietterich (2019); Jiang & Lin (2023), randomly apply 15 different kinds of corruptions to the source clients, and 4 new kinds of corruptions to the target clients to test the generalization of ATP. For label shift, we use the step partition Chen & Chao (2020), where each client has 8 minor classes with 5 images per class, and 2 major classes with 80 images

per class. For the hybrid shift, we apply both step partition and feature perturbations simultaneously on each client. We also test DynFed with two different architectures: a five-layer CNN on CIFAR-10(-C) and ResNet-18 on CIFAR-100(-C). Baseline models we chose several common TTA models: BN-adapt Schneider et al. (2020), Tent Wang et al. (2021a), T3A Iwasawa & Matsuo (2021), MEMO Zhang et al. (2022) and the current state-of-the-art TTPFL framework ATP Bao et al. (2024). To simplify the expression, we use DynFed-B and DynFed-O to denote the TTBA and OTTA variants, respectively.

## 5.2 Experiment results

### 5.2.1 Performance comparison

Table 1: Accuracy (%) on target clients under various distribution shifts on CIFAR-10 (CNN) and CIFAR-100 (ResNet-18). Best results in **bold**.

| Method | CIFAR-10 | | | | CIFAR-100 | | | |
|---|---|---|---|---|---|---|---|---|
| | Feature | Label | Hybrid | MT | Feature | Label | Hybrid | MT |
| No adapt | 64.36±.13 | 69.42±.24 | 63.68±.24 | 57.71±.20 | 35.12±.22 | 38.88±.32 | 33.68±.30 | 26.50±.24 |
| BN adapt | 65.52±.22 | 64.54±.10 | 60.02±.39 | 54.02±.18 | 36.52±.27 | 35.54±.15 | 32.02±.45 | 25.80±.22 |
| Tent | 65.76±.09 | 70.13±.21 | 63.42±.26 | 57.62±.24 | 36.76±.13 | 39.13±.28 | 34.05±.33 | 27.30±.28 |
| T3A | 64.53±.08 | 71.70±.32 | 62.17±.17 | 58.48±.21 | 35.53±.12 | 40.70±.38 | 33.17±.23 | 28.00±.25 |
| MEMO | 62.43±.22 | 78.45±.15 | 63.07±.29 | 55.49±.12 | 34.43±.30 | 44.31±.29 | 34.07±.37 | 26.30±.17 |
| ATP Bao et al. (2024) | 66.54±.12 | 76.67±.22 | 68.37±.35 | 60.69±.19 | 39.54±.27 | 48.97±.30 | 40.12±.40 | 31.19±.25 |
| DynFed-B | **68.12±.18** | **78.54±.19** | **69.56±.31** | **62.23±.17** | **40.49±.22** | **50.19±.25** | **41.18±.35** | **33.72±.20** |
| DynFed-O | **68.49±.25** | **79.18±.27** | **69.67±.36** | **62.47±.19** | **41.17±.28** | **51.02±.32** | **41.48±.38** | **34.05±.22** |

We evaluate DynFed against state-of-the-art test-time adaptation techniques on CIFAR-10 and CIFAR-100 datasets under various distribution shifts. The results are shown in Table 1. Our method consistently outperforms all baselines, including the current state-of-the-art TTPFL method ATP, across all shift types. On CIFAR-10, DynFed-O achieves accuracy improvements of 1.95%, 2.51%, 1.30%, and 1.78% over ATP for feature, label, hybrid, and multi-type shifts, respectively. The gains are even more significant on CIFAR-100, with improvements of 1.63%, 2.05%, 1.36%, and 2.86% for the same shift types. Both evaluation variants of our method consistently outperform other approaches, with the OTTA variant (DynFed-O) showing slightly better results, particularly in complex shift scenarios like multi-type shifts. These results demonstrate DynFed's effectiveness in handling various distribution shifts in FL environments. The substantial improvements over ATP, especially in challenging scenarios like multi-type shifts and on the more complex CIFAR-100 dataset, highlight our method's robustness and flexibility stemming from the dynamic, client-specific adaptation rates generated by the ARN. DynFed not only advances the state-of-the-art in test-time adaptation for FL but also shows remarkable consistency across different shift types, underscoring its potential for real-world applications with unknown or mixed distribution shifts.

### 5.2.2 Hyperparameter analysis

In the ARN component (Eq. 10), there is a critical hyperparameter $\tau$, the scaling factor. Our experiments revealed that the optimal value of $\tau$ depends significantly on the nature of the distribution shift, aligning with our theoretical insight regarding the dichotomy between one-type and multi-type shifts (Theorem 3.3). Figure 3 illustrates the crucial role of $\tau$ in our ARN across different distribution shift scenarios on CIFAR-10. For multi-type distribution shifts (red line), we observe optimal performance around $\tau = 0.3$, with accuracy peaking at 62.33%. The performance rapidly declines as $\tau$ increases beyond this point, stabilizing at a lower level for $\tau > 1$. Conversely, for one-type distribution shifts, exemplified by feature shift (blue line), the model's accuracy improves as $\tau$ increases, reaching its optimal value of 68.27% around $\tau = 1.7$. This performance remains relatively stable for higher $\tau$ values. This contrasting behavior underscores the adaptive capability required to handle diverse distribution shift scenarios. While $\tau$ is a hyperparameter in our current implementation, this finding suggests potential future work in learning $\tau$ dynamically. The significant performance differences highlight the importance of correctly scaling the adaptation rates generated by the ARN, especially in TTPFL environments with heterogeneous distribution shifts.

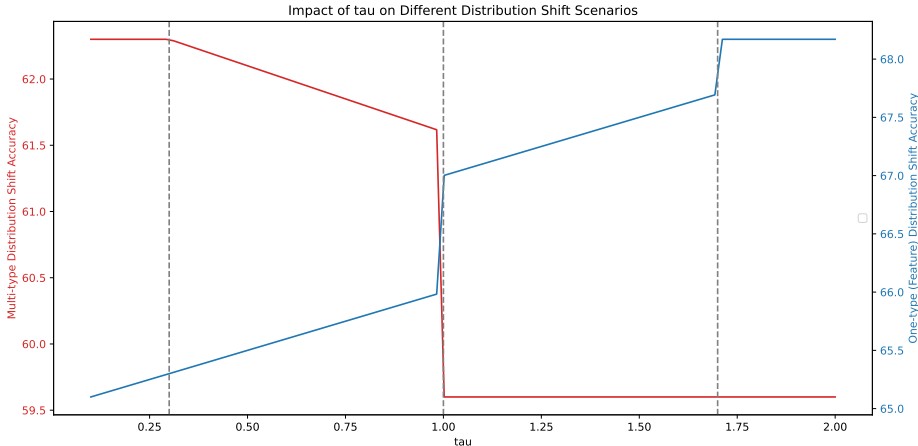

Figure 3: Impact of ARN scaling factor $\tau$ on model accuracy under different distribution shift scenarios (CIFAR-10). The red line represents accuracy for multi-type distribution shift, while the blue line shows accuracy for one-type distribution shift (feature shift). Optimal $\tau$ values differ significantly.

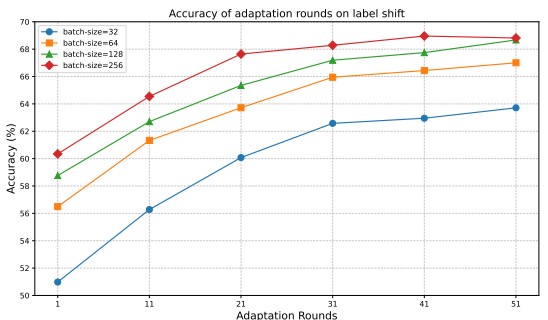

(a) Impact of batch size on accuracy over adaptation rounds (Label Shift, CIFAR-10).

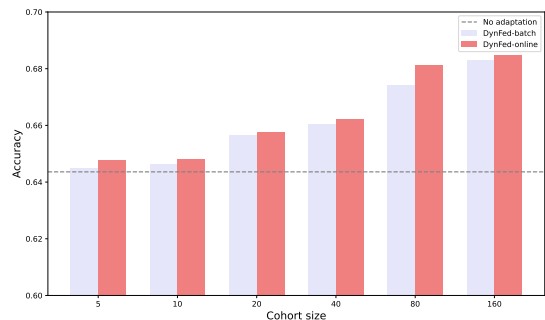

(b) Impact of shard size (total data per client) on accuracy (Feature Shift, CIFAR-10).

Figure 4: Effect of batch size and shard size on DynFed performance.

### 5.2.3 Effect of batch size and shard size

Figure 4 illustrates the impact of batch size and shard size (total amount of test data available per client) on our method's performance under different distribution shift scenarios on CIFAR-10. As shown in Figure 4a, larger batch sizes generally yield higher accuracy in label shift conditions during the initial adaptation rounds, with batch size 256 achieving the best early performance. However, the performance gap between batch sizes narrows over time, suggesting diminishing returns for larger batches in later adaptation rounds, possibly because smaller batches allow for more frequent adaptation steps. Figure 4b demonstrates that both DynFed variants significantly outperform the no-adaptation baseline across all shard sizes in feature shift scenarios. DynFed-O shows superior performance compared to DynFed-B, especially as the shard size increases, likely benefiting from the accumulated information in the averaged update direction ($\bar{h}_k^T$ in Eq. 12). These results highlight our method's adaptability and efficiency across various operational conditions, showcasing its robustness in diverse federated learning environments and its ability to effectively leverage available data for improved adaptation.

### 5.2.4 Optimization trajectories

Figure 5 provides a qualitative visualization of the adaptation dynamics of our proposed method (DynFed) compared to a baseline TTPFL method (ATP) during the testing phase on target clients. The figure

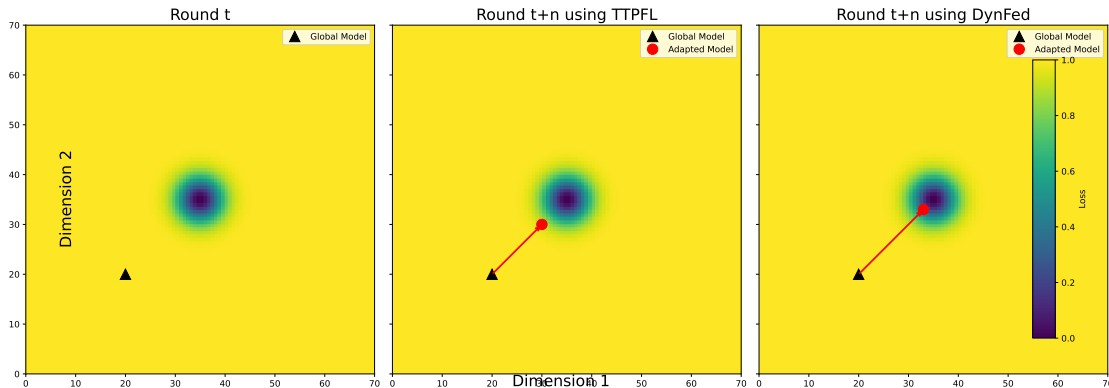

Figure 5: Optimization trajectories of TTPFL (ATP) and DynFed in FL, projected onto two principal dimensions of adaptation rate changes during testing on target clients. DynFed demonstrates convergence towards a potentially better region (darker area indicates lower loss) compared to TTPFL, illustrating its more effective adaptation dynamics.

shows trajectories in a 2D space obtained by projecting the changes in adaptation rates (or adapted model parameters) onto the principal components of variation. Both methods start from the same initial state (corresponding to the global model $\mathbf{w}_G$). Over $n = 50$ adaptation rounds, DynFed appears to navigate the adaptation landscape more effectively, moving towards regions associated with lower loss (darker area) more consistently than the baseline. This visual representation suggests that DynFed's dynamic rate generation mechanism allows for a more refined and potentially faster convergence towards a better-adapted state for the target client's specific data distribution.

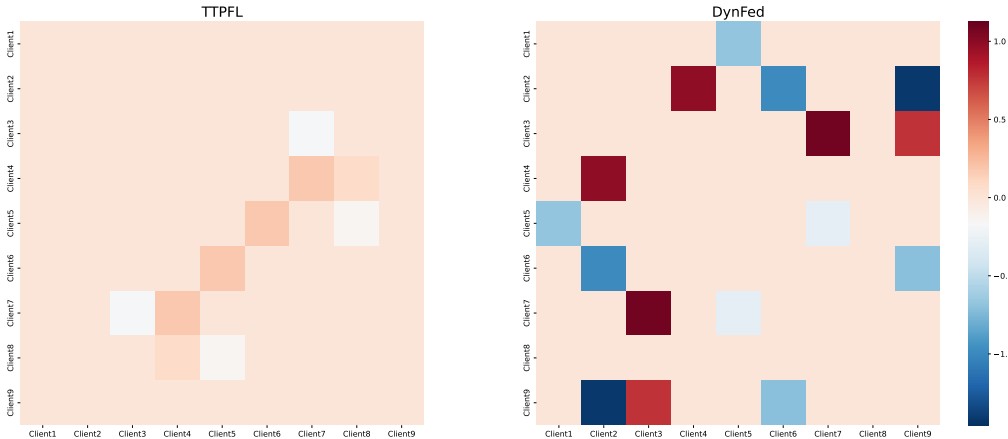

Figure 6: Heatmap comparison of learned adaptation rates ($\boldsymbol{\alpha}$) across different model modules (y-axis) for different clients (x-axis) under a multi-type shift scenario. Left: TTPFL (ATP, using uniform rates). Right: DynFed (showing diverse, client- and module-specific rates).

### 5.2.5 Performance on Domain Generalization Benchmarks

To further assess DynFed's robustness, we evaluated it on two standard domain generalization benchmarks, Digits-5 and PACS, under a hybrid shift scenario (combining feature and label shifts). Digits-5 comprises five digit recognition domains (MNIST, SVHN, USPS, SynthDigits, MNIST-M), while PACS includes four object recognition domains (Art, Cartoon, Photo, Sketch). We followed the standard leave-one-domain-out evaluation protocol, where one domain serves as the unseen target domain for testing, while the remaining domains act as source domains for training the initial global model and the adaptation mechanism (ARN

Table 2: Accuracy (%) on target clients under *hybrid shift* on Digits-5 and PACS domain generalization benchmarks. The source model is ResNet-18 pretrained on ImageNet.

| 2*Method | Digits-5 | | | | | PACS | | | |
|---|---|---|---|---|---|---|---|---|---|
| | MNIST | SVHN | USPS | SynthD | MNIST-M | Art | Cartoon | Photo | Sketch |
| No adapt | 95.5±0.2 | 52.3±1.5 | 89.6±0.4 | 79.8±0.7 | 55.6±0.8 | 71.6±1.2 | 74.7±0.7 | 90.3±0.8 | 74.2±0.7 |
| BN-Adapt | 94.9±0.3 | 57.6±0.5 | 89.5±0.4 | 75.3±0.5 | 59.7±0.4 | 73.6±0.5 | 71.5±0.6 | 92.1±0.3 | 70.9±0.5 |
| SHOT | 94.7±0.3 | 57.9±0.2 | 89.6±0.7 | 76.4±0.3 | 60.2±0.7 | 69.3±0.7 | 67.8±0.4 | 87.0±0.6 | 59.4±0.9 |
| Tent | 95.5±0.3 | 60.7±0.5 | 91.7±0.6 | 78.6±0.5 | 62.5±0.7 | 71.6±0.7 | 71.0±1.0 | 88.1±0.2 | 63.2±1.1 |
| T3A | 94.6±0.6 | 49.9±1.1 | 88.5±0.8 | 75.5±1.1 | 51.3±1.6 | 72.2±0.7 | 75.0±0.8 | 91.5±0.6 | 70.1±1.2 |
| MEMO | 95.9±0.2 | 52.9±1.1 | 89.8±0.4 | 80.1±0.9 | 55.5±1.1 | 71.5±1.3 | 75.6±1.0 | 90.7±0.9 | 76.3±0.7 |
| EM | 96.6±0.3 | 57.2±1.7 | 92.3±0.3 | 85.7±0.5 | 62.1±0.6 | 74.0±1.9 | 78.9±0.9 | 92.3±0.9 | 80.8±1.5 |
| BBSE | 94.5±0.6 | 57.3±1.5 | 91.3±0.4 | 85.5±0.5 | 61.6±0.9 | 74.3±1.8 | 78.7±1.0 | 91.8±0.7 | 80.2±1.4 |
| Surgical | 97.4±0.1 | 59.9±2.0 | 94.2±0.4 | 86.1±0.4 | 65.9±0.8 | 74.6±2.7 | 77.5±0.6 | 92.3±0.8 | 80.9±3.4 |
| ATP Bao et al. (2024) | 97.8±0.3 | **62.2±1.7** | 95.4±0.3 | **87.9±0.5** | 70.0±2.0 | **82.9±1.0** | 79.6±0.8 | 95.4±0.4 | 82.3±1.6 |
| DynFed-B | 97.7±0.2 | 61.9±1.7 | **95.6±0.3** | 87.8±0.5 | 68.9±1.8 | 81.9±1.1 | **80.1±0.7** | 94.9±0.4 | 82.0±1.6 |
| DynFed-O | **97.9±0.2** | 61.7±1.8 | 95.4±0.3 | 86.4±0.5 | **71.1±2.1** | 82.8±0.9 | 79.8±0.7 | **95.7±0.4** | **82.6±1.5** |

and initial rates). Table **??** presents the results using a ResNet-18 backbone pretrained on ImageNet. Our DynFed method demonstrates strong performance across most domains, often achieving results comparable to or exceeding the state-of-the-art ATP method. DynFed-B achieves the best accuracy on USPS and Cartoon domains, while DynFed-O excels on MNIST, MNIST-M, Photo, and Sketch domains. Notably, DynFed-O shows consistent improvements over DynFed-B across multiple target domains (SVHN, MNIST-M, Photo, Sketch), suggesting the benefit of the online averaging mechanism (Eq. 12) in these challenging adaptation scenarios. These results highlight DynFed's effectiveness in adapting to complex real-world domain shifts, where it either matches or outperforms state-of-the-art methods, further validating the utility of its dynamic adaptation approach.

### 5.2.6 Adaptive flexibility in client-specific batch data handling

To verify that our method can efficiently adapt to different types of data distributions across clients, we examined the learned adaptation rates in a multi-type shift scenario. Figure 6 visualizes the adaptation rates ($\boldsymbol{\alpha}$) learned for different model modules (y-axis) across 9 different clients (x-axis). The TTPFL method (ATP, left panel) uses uniform rates, resulting in identical heatmap columns for all clients, indicating no client-specific customization. In contrast, our method (DynFed, right panel) exhibits significant variation in adaptation rates both across clients (different columns) and across modules within a client (different rows within a column). This diversity, indicated by the varied color intensities, demonstrates DynFed's ability to generate highly personalized adaptation strategies tailored to each client's unique data characteristics and the specific needs of different model parts (e.g., potentially higher rates for BN layers for a client experiencing feature shift, higher rates for the final layer for a client with label shift). This adaptive behavior is crucial for effectively handling the diverse data distributions encountered in real-world federated learning scenarios.

To further validate this client-specific effectiveness, we compared the accuracy of DynFed and TTPFL (ATP) on 10 randomly sampled target clients from the CIFAR-10 multi-type shift distribution. Figure 7 illustrates the performance for each client. As shown in the bar chart, DynFed (blue bars) consistently outperforms TTPFL (orange bars) across all individual clients. Furthermore, the average accuracy of DynFed across these clients (blue dashed line) is notably higher than that of TTPFL (orange dashed line). This comprehensive superiority, both in individual client performance and overall average accuracy, underscores the adaptability and effectiveness of our method in diverse federated learning scenarios.

Table 3: Ablation study: accuracy (%) on target clients under various distribution shifts (CIFAR-10, CNN). 'w/o opt. $\tau$' uses a suboptimal $\tau$ (e.g., $\tau = 1.7$ for MT shift, $\tau = 0.3$ for Feature shift).

| Method | Feature Shift | Hybrid Shift | MT Shift |
|---|---|---|---|
| No adapt | 64.36±0.13 | 63.68±0.24 | 57.71±0.20 |
| DynFed w/o ARN (uses $\boldsymbol{\alpha}_G$) | 66.71±0.27 | 68.31±0.36 | 60.62±0.19 |
| DynFed w/o opt. $\tau$ | 65.23±0.23 | 65.69±0.37 | 57.75±0.21 |
| DynFed-B (Full) | **68.12±0.18** | **69.56±0.31** | **62.23±0.17** |

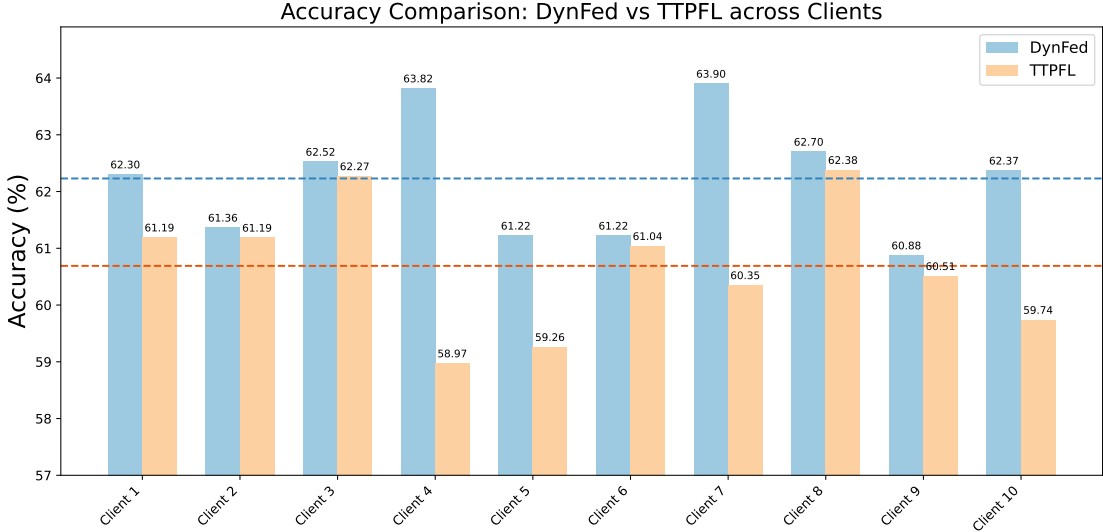

Figure 7: Accuracy comparison between TTPFL (ATP) and DynFed on 10 individual target clients under multi-type shift (CIFAR-10). Dashed lines represent average accuracy across these clients.

### 5.2.7 Ablation study

Ablation studies were conducted on CIFAR-10 (CNN) to evaluate the contribution of key components in our method, as shown in Table **??**. First, removing the ARN entirely and using only the aggregated initial rates $\boldsymbol{\alpha}_G$ (similar to ATP but potentially with learned initial rates) results in the 'DynFed w/o ARN' variant. This performs better than 'No adapt' but worse than the full DynFed-B, highlighting the benefit of dynamic rate adjustment during testing provided by the ARN. Second, we tested the impact of the ARN scaling factor $\tau$. The 'DynFed w/o opt. $\tau$' variant uses the ARN but with a $\tau$ value that is optimal for the *opposite* type of shift (e.g., using $\tau \approx 1.7$ for MT shift, based on Fig. 3). This leads to significantly lower performance, even worse than 'DynFed w/o ARN' in the MT shift scenario, where accuracy drops to 57.75%, barely surpassing the 'No adapt' baseline. This underscores the critical role of $\tau$ in effectively leveraging the ARN's capabilities and aligns with Theorem 3.3, showing that applying the wrong adaptation scaling can be detrimental. The full DynFed-B method, incorporating both the ARN and an appropriately chosen $\tau$ (tuned on a validation set or based on the expected shift type), consistently outperforms all ablated variants. This demonstrates the synergistic effect of these components in handling various unseen distribution shifts, with the most significant improvements observed in the challenging hybrid and multi-type distribution shift scenarios.

### 5.3 Limitations and Potential Failure Cases

While DynFed demonstrates superior performance across various distribution shift scenarios, it is important to acknowledge its limitations and potential failure cases:

### 5.3.1 Extreme Heterogeneity and Shift Complexity

In scenarios with extremely high degrees of heterogeneity where each client experiences a unique and complex combination of multiple distribution shift types, the ARN, trained on potentially simpler source client shifts, may struggle to generate optimal adaptation rates. The adaptation strategy dichotomy (Theorem 3.3) suggests that when faced with highly diverse or unseen shift combinations, the ARN might converge to compromise solutions that are suboptimal for some individual clients. Furthermore, the effectiveness relies on the expressiveness of the ARN architecture and the quality of the initial rates learned during training.

### 5.3.2 Computational Overhead

DynFed introduces additional computational complexity compared to fixed-rate TTPFL methods like ATP or standard TTA methods. During testing (Algorithm 2), each adaptation step requires a forward pass through the ARN (Eq. 11) to compute $\boldsymbol{\alpha}_{\text{current}}$, in addition to the gradient computation ($\mathbf{h}_{jk}^T$) and model adaptation step. While the ARN is designed to be compact, this overhead might be non-negligible for resource-constrained edge devices (e.g., IoT devices, low-power mobile phones) with limited computational capabilities or strict latency requirements. The training phase (Algorithm 1) also involves computing gradients with respect to the adaptation rates $\boldsymbol{\alpha}$ (Eq. 9), adding to the local computation cost on source clients. The trade-off between adaptation performance and computational cost needs careful consideration for specific applications.

### 5.3.3 Hyperparameter Sensitivity

As shown in Figure 3 and Table **??**, the performance of DynFed can be sensitive to the choice of the ARN scaling factor $\tau$. While we demonstrated that different shift types require different optimal $\tau$ values, selecting the best $\tau$ for unseen target clients remains a challenge. In our experiments, $\tau$ was treated as a hyperparameter tuned based on the expected shift type or a validation set. An improperly chosen $\tau$ can significantly degrade performance. Future work could explore methods for dynamically estimating or adapting $\tau$ itself during test time. The momentum parameter $\lambda$ (Eq. 11) also requires tuning.

### 5.4 Conclusion

In this paper, we introduced DynFed, a novel approach for test-time adaptation in federated learning that effectively addresses the challenges of heterogeneous distribution shifts across clients. Our method leverages an Adaptive Rate Network (ARN) function with an optimized scaling factor $\tau$ to generate dynamic, client-specific, and module-specific adaptation rates. This allows DynFed to navigate the fundamental dichotomy between optimal strategies for different shift types. Through extensive experiments on various benchmarks, including standard FL datasets and domain generalization tasks, we demonstrated that DynFed significantly outperforms existing TTA and TTPFL methods across diverse shift scenarios, particularly in complex multi-type shift environments. Our theoretical analysis provides convergence and generalization guarantees, supporting the stability of our approach, while propositions justify the need for adaptive, component-specific rates. Ablation studies confirmed the crucial roles of both the ARN and the appropriate scaling factor $\tau$. Despite limitations related to computational overhead and hyperparameter sensitivity, DynFed offers a robust and adaptive solution, paving the way for more effective federated learning in real-world applications where distribution shifts are diverse and unpredictable.

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

# A    Appendix: Theoretical Analysis

In this section, we provide theoretical justification for DynFed by analyzing its convergence properties and generalization guarantees. We show that despite having client-specific adaptation mechanisms, DynFed maintains both convergence and generalization capabilities comparable to traditional federated learning methods. Moreover, we demonstrate why different adaptation strategies are necessary for different types of distribution shifts.

## A.1    Convergence Analysis

We first analyze the convergence properties of the DynFed training process (Algorithm 1) to demonstrate that our dynamic adaptation approach converges despite the additional complexity of learning client-specific, adaptive rates. Similar to previous work in federated learning McMahan et al. (2017), we establish convergence guarantees under standard assumptions. In particular, we consider the objective function of DynFed training as minimizing the expected loss over the source client population after adaptation:

$$\mathcal{L}(\mathbf{w}_G, \{\boldsymbol{\alpha}_j\}_{j=1}^M) = \frac{1}{M} \sum_{j=1}^M \mathbb{E}_{X_k, Y_k \sim P_j} [\ell_{CE}(f(X_k; \mathbf{w}_G + A(\boldsymbol{\alpha}_j) \odot \mathbf{h}_k(X_k, \mathbf{w}_G)), Y_k)], \tag{16}$$

where $\mathbf{w}_G$ represents the global model parameters and $\{\boldsymbol{\alpha}_j\}_{j=1}^M$ denotes the set of adaptation rate vectors for $M$ source clients. Here, $\mathcal{L}_{P_j}(\mathbf{w}_G, \boldsymbol{\alpha}_j)$ is the expected loss for client $j$ after one step of adaptation using rates $\boldsymbol{\alpha}_j$. The actual algorithm involves multiple local steps and aggregation, which we analyze in expectation.

To analyze convergence, we define the following Lyapunov function that tracks the distance from the current parameters (global model and average rates) to the optimal ones $(\mathbf{w}_G^*, \boldsymbol{\alpha}^*)$:

$$V_t = \|\mathbf{w}_G^t - \mathbf{w}_G^*\|^2 + \gamma M \|\bar{\boldsymbol{\alpha}}^t - \boldsymbol{\alpha}^*\|^2, \tag{17}$$

where $\bar{\boldsymbol{\alpha}}^t = \frac{1}{M} \sum_j \boldsymbol{\alpha}_j^t$ is the average adaptation rate vector across clients at round $t$, $\boldsymbol{\alpha}^*$ is the optimal average rate vector, and $\gamma > 0$ is a constant balancing the two terms. (Note: This analysis focuses on the convergence of the average rates; convergence of individual $\boldsymbol{\alpha}_j$ might require stronger assumptions or different analysis).

[Proof of Theorem 4.4.1] We provide a sketch based on standard FL convergence analysis, adapted for the joint optimization of $\mathbf{w}_G$ and $\boldsymbol{\alpha}$. Let $\mathbf{w}_G^{t+1}$ and $\boldsymbol{\alpha}_j^{t+1}$ be the parameters after round $t + 1$. The update involves local steps and aggregation. We analyze the expected change in the Lyapunov function over one round.

Let $\mathcal{L}^t = \mathcal{L}(\mathbf{w}_G^t, \{\boldsymbol{\alpha}_j^t\}_{j=1}^M)$. The expected update relies on stochastic gradients $\hat{\nabla}_{\mathbf{w}_G}^t$ and $\hat{\nabla}_{\boldsymbol{\alpha}_j}^t$ obtained from sampled clients and batches. We assume these gradients are unbiased estimates of the true gradients of $\mathcal{L}^t$ in expectation over the sampling process.

Consider the change in the Lyapunov function $V_t$. Using standard descent lemma arguments under $\beta$-smoothness (Assumption 1) and convexity (Assumption 3), and accounting for the variance of stochastic gradients (Assumption 2), we can bound the expected progress per round:

$$\mathbb{E}[V_{t+1}|\mathcal{F}_t] - V_t \leq -\eta_w \langle \nabla_{\mathbf{w}_G} \mathcal{L}^t, \mathbf{w}_G^t - \mathbf{w}_G^* \rangle + \frac{\beta \eta_w^2}{2} \mathbb{E}[\|\hat{\nabla}_{\mathbf{w}_G}^t\|^2 | \mathcal{F}_t]$$
$$- \eta_\alpha \gamma \sum_{j \in S_t} \langle \nabla_{\boldsymbol{\alpha}_j} \mathcal{L}^t, \boldsymbol{\alpha}_j^t - \boldsymbol{\alpha}_j^* \rangle + \frac{\beta \eta_\alpha^2 \gamma}{2} \sum_{j \in S_t} \mathbb{E}[\|\hat{\nabla}_{\boldsymbol{\alpha}_j}^t\|^2 | \mathcal{F}_t] + \text{FL terms}$$

where $\mathcal{F}_t$ is the history up to round $t$, $S_t$ is the set of clients sampled at round $t$, and 'FL terms' capture effects of local updates and averaging which can introduce gradient drift, bounded under standard FL analysis assumptions (e.g., bounded gradient dissimilarity).

Using convexity ($\langle \nabla f(x), x - x^* \rangle \geq f(x) - f(x^*) \geq \frac{1}{2\beta} \|\nabla f(x)\|^2$ for $\beta$-smooth convex $f$), the bounded variance assumption ($\mathbb{E}[\|\hat{\nabla}\|^2] \leq \|\nabla\|^2 + \sigma^2$), and choosing appropriate learning rates ($\eta_w, \eta_\alpha \leq 1/\beta$), we get:

$$\mathbb{E}[V_{t+1}|\mathcal{F}_t] - V_t \leq -\frac{\eta_w}{2}\|\nabla_{\mathbf{w}_G}\mathcal{L}^t\|^2 - \frac{\gamma\eta_\alpha}{2}\mathbb{E}\left[\sum_{j \in S_t}\|\nabla_{\boldsymbol{\alpha}_j}\mathcal{L}^t\|^2|\mathcal{F}_t\right]$$
$$+ \frac{\eta_w^2\beta}{2}\sigma_w^2 + \frac{\gamma\eta_\alpha^2\beta}{2}\sigma_\alpha^2 + \text{FL error terms}$$

Taking total expectation and summing over $t = 0$ to $T - 1$:

$$\mathbb{E}[V_T] - \mathbb{E}[V_0] \leq -\sum_{t=0}^{T-1}\mathbb{E}\left[\frac{\eta_w}{2}\|\nabla_{\mathbf{w}_G}\mathcal{L}^t\|^2 + \frac{\gamma\eta_\alpha}{2M}\sum_{j=1}^{M}\|\nabla_{\boldsymbol{\alpha}_j}\mathcal{L}^t\|^2\right]$$
$$+ T\left(\frac{\eta_w^2\beta}{2}\sigma_w^2 + \frac{\gamma\eta_\alpha^2\beta}{2}\sigma_\alpha^2\right) + \sum_{t=0}^{T-1}\mathbb{E}[\text{FL error terms}]$$

Rearranging gives the bound on the average squared gradient norm as stated in Theorem 4.4.1, assuming the FL error terms are appropriately bounded or decay. This shows convergence to a neighborhood determined by variances and learning rates.

## A.2 Generalization Guarantees

Next, we analyze the generalization properties of the learned adaptation strategy (represented by the initial rates $\boldsymbol{\alpha}_G$ and the ARN $g$). We want to bound the difference between the expected error on unseen target clients and the empirical error observed on source clients during training.

[Proof of Theorem 4.4.2] We use Rademacher complexity bounds to analyze the generalization error. Let $\mathcal{F}_\mathcal{H}$ be the class of functions mapping input data $X$ to post-adaptation predictions, parameterized by $\boldsymbol{\alpha} \in \mathcal{H} = \{\boldsymbol{\alpha} : \|\boldsymbol{\alpha}\|_2 \leq R\}$. The prediction for input $x$ using rate $\boldsymbol{\alpha}$ is $f(x; \mathbf{w}_G + A(\boldsymbol{\alpha}) \odot \mathbf{h}(x, \mathbf{w}_G))$. Let $\ell$ be the 0-1 loss. We want to bound $\sup_{\boldsymbol{\alpha} \in \mathcal{H}} |\mathbb{E}[\ell(f_{\boldsymbol{\alpha}}(X), Y)] - \hat{\mathbb{E}}[\ell(f_{\boldsymbol{\alpha}}(X), Y)]|$.

Standard learning theory results (e.g., Theorem 3.3 in **?**) state that with probability at least $1 - \delta$:

$$\sup_{\boldsymbol{\alpha} \in \mathcal{H}} |\varepsilon(\boldsymbol{\alpha}) - \hat{\varepsilon}(\boldsymbol{\alpha})| \leq 2\mathfrak{R}_{NK}(\ell \circ \mathcal{F}_\mathcal{H}) + \sqrt{\frac{\log(1/\delta)}{2NK}}$$

where $\mathfrak{R}_{NK}$ is the empirical Rademacher complexity over the $NK$ source samples.

We need to bound the Rademacher complexity $\mathfrak{R}_{NK}(\ell \circ \mathcal{F}_\mathcal{H})$. Since the 0-1 loss is bounded, by Talagrand's contraction lemma, $\mathfrak{R}_{NK}(\ell \circ \mathcal{F}_\mathcal{H}) \leq 2\mathfrak{R}_{NK}(\mathcal{F}_\mathcal{H})$. We bound the complexity of the function class $\mathcal{F}_\mathcal{H}$. Assuming the model $f(x; \mathbf{w})$ is $L_w$-Lipschitz w.r.t. parameters $\mathbf{w}$, and the adaptation $A(\boldsymbol{\alpha}) \odot \mathbf{h}$ changes parameters by at most $\|\boldsymbol{\alpha}\|\|\mathbf{h}\| \leq RHd^{1/2}$ (assuming elementwise product and bounded $h^{[l]}$), the function $f_{\boldsymbol{\alpha}}(x)$ is related to $f_{\mathbf{0}}(x) = f(x; \mathbf{w}_G)$. The complexity depends on how changes in $\boldsymbol{\alpha}$ affect the output $f_{\boldsymbol{\alpha}}(x)$. Using Lipschitz properties and the structure of the adaptation: $\|f_{\boldsymbol{\alpha}_1}(x) - f_{\boldsymbol{\alpha}_2}(x)\| \leq L_w\|(A(\boldsymbol{\alpha}_1) - A(\boldsymbol{\alpha}_2)) \odot \mathbf{h}\| \leq L_wHd^{1/2}\|\boldsymbol{\alpha}_1 - \boldsymbol{\alpha}_2\|$. This suggests the function class $\mathcal{F}_\mathcal{H}$ has a complexity related to the complexity of the parameter space $\mathcal{H}$.

Using bounds based on covering numbers (similar to the sketch in the main text but with Rademacher complexity): The Rademacher complexity of a function class whose outputs are bounded and depend Lipschitz-continuously on parameters in a $d$-dimensional ball of radius $R$ is often bounded by $\mathcal{O}(\sqrt{d\log(R)/NK})$. Combining these yields a bound of the form:

$$\sup |\varepsilon - \hat{\varepsilon}| \leq \mathcal{O}\left(\sqrt{\frac{d\log(R)}{NK}} + \sqrt{\frac{\log(1/\delta)}{NK}}\right)$$

The exponential dependency in Theorem 4.4.2 likely arises from using covering numbers directly with Hoeffding/union bound, which can yield tighter constants but potentially looser dependency on $d$ in the pre-factor. The exact form depends on the specific assumptions and proof technique (covering numbers vs. Rademacher complexity). The key takeaway is that the generalization error decreases with $N$ and $K$, and depends on the complexity $(d, R)$ of the adaptation space.

### A.3   Adaptive Mechanisms for Different Distribution Shifts

Here, we provide the proofs for the propositions motivating different adaptation strategies.

[Proof of Proposition 4.4.3] Let $f(\mathbf{x}; \mathbf{w})$ be the network outputting logits $z_c = \mathbf{w}_c^\top g(\mathbf{x}) + b_c$ for class $c$, where $g(\mathbf{x})$ are features from preceding layers. The softmax output is $p(y = c|\mathbf{x}) = \frac{\exp(z_c)}{\sum_{c'} \exp(z_{c'})}$. The network is calibrated on $p$, so $p(y = c|\mathbf{x})$ matches the true conditional probability under distribution $p$. We want to find parameters $\mathbf{w}'$ such that the network output matches $q(y = c|\mathbf{x})$. By Bayes' theorem under label shift $(p(\mathbf{x}|y) = q(\mathbf{x}|y))$:

$$
\begin{aligned}
q(y = c|\mathbf{x}) &= \frac{q(\mathbf{x}|y = c)q(y = c)}{q(\mathbf{x})} \\
&= \frac{p(\mathbf{x}|y = c)q(y = c)}{q(\mathbf{x})} \\
&= \frac{p(y = c|\mathbf{x})p(\mathbf{x})}{p(y = c)} \frac{q(y = c)}{q(\mathbf{x})} \\
&= p(y = c|\mathbf{x}) \frac{p(\mathbf{x})}{q(\mathbf{x})} \frac{q(y = c)}{p(y = c)}
\end{aligned}
$$

The term $\frac{p(\mathbf{x})}{q(\mathbf{x})}$ is constant across classes $c$ for a given $\mathbf{x}$. Let $r_c = q(y = c)/p(y = c)$. So, $q(y = c|\mathbf{x}) \propto p(y = c|\mathbf{x})r_c = \frac{\exp(z_c)}{\sum_{c'} \exp(z_{c'})}r_c$. We want the new network output $q'(y = c|\mathbf{x}) = \frac{\exp(z_c')}{\sum_{c'} \exp(z_{c'}')}$ to equal $q(y = c|\mathbf{x})$. This requires $\exp(z_c') \propto \exp(z_c)r_c$, which means $z_c' = z_c + \log r_c + \text{const}$. Setting $z_c' = z_c + \log r_c = z_c + \log(q(y = c)/p(y = c))$ achieves this. Since $z_c = \mathbf{w}_c^\top g(\mathbf{x}) + b_c$, the new logit $z_c'$ corresponds to keeping $\mathbf{w}_c$ and $g(\mathbf{x})$ the same, but changing the bias to $b_c' = b_c + \log(q(y = c)/p(y = c))$. Thus, adapting only the final layer bias is sufficient.

[Proof Sketch of Proposition 4.4.3] Feature shift implies $p(\mathbf{x}) \neq q(\mathbf{x})$ but $p(y|\mathbf{x}) = q(y|\mathbf{x})$. If the shift primarily affects the low-order statistics (mean $\mu$, variance $\sigma^2$) of activations $\mathbf{z}$ at some layer, BN aims to normalize these statistics. BN transforms $z$ to $\hat{z} = (z - \mu_{\text{batch}})/\sqrt{\sigma_{\text{batch}}^2 + \epsilon}$, and then scales and shifts it: $\gamma\hat{z} + \beta_{\text{BN}}$. During inference, it uses running statistics $\mu_{\text{run}}, \sigma_{\text{run}}^2$. If the target distribution $q$ has different activation statistics $\mu_q, \sigma_q^2$ compared to the source $p$ statistics $\mu_p, \sigma_p^2$ (which $\mu_{\text{run}}, \sigma_{\text{run}}^2$ approximate), the normalization will be incorrect for target data. Adapting the BN layer involves updating $\mu_{\text{run}}, \sigma_{\text{run}}^2$ towards the target statistics $\mu_q, \sigma_q^2$. If the shift is primarily captured by these first and second moments, adapting the BN statistics can effectively reverse the shift's impact on the normalized activations passed to subsequent layers, thus restoring model performance without changing the weights of convolutional or linear layers. This is the principle behind methods like BN-adapt.

### A.4   Adaptation Strategy Dichotomy

We provide a more formal argument for Theorem 3.3.

[Proof of Theorem 3.3] Let $\mathcal{L}(\boldsymbol{\alpha})$ be the post-adaptation loss function, assumed to be strongly convex. The optimal adaptation rate vector $\boldsymbol{\alpha}^*$ satisfies $\nabla_{\boldsymbol{\alpha}}\mathcal{L}(\boldsymbol{\alpha}^*) = 0$. Consider the one-type shift scenario $\mathcal{S}$. Let the loss be $\mathcal{L}_{\mathcal{S}}(\boldsymbol{\alpha}) = \frac{1}{N}\sum_{i=1}^{N}\mathcal{L}_{\mathcal{S}_i}(\boldsymbol{\alpha})$. Assume the shift requires adaptation primarily in a subspace spanned by direction vector $\mathbf{v}_{\mathcal{S}}$ (e.g., adapting BN layers for feature shift, or last layer for label shift). Then, the gradient $\nabla_{\boldsymbol{\alpha}}\mathcal{L}_{\mathcal{S}}(\boldsymbol{\alpha})$ will be roughly aligned with $-\mathbf{v}_{\mathcal{S}}$ when $\boldsymbol{\alpha}$ is small. The optimum $\boldsymbol{\alpha}_{\mathcal{S}}^*$ will likely be in the direction of $\mathbf{v}_{\mathcal{S}}$.

Consider the multi-type shift scenario $\mathcal{M}$. Let the loss be $\mathcal{L}_\mathcal{M}(\boldsymbol{\alpha}) = \frac{1}{N}\sum_{i=1}^{N}\mathcal{L}_{\mathcal{M}_i}(\boldsymbol{\alpha})$. Suppose half the clients ($\mathcal{M}_{\text{feat}}$) have feature shift requiring adaptation in direction $\mathbf{v}_{\text{feat}}$, and half ($\mathcal{M}_{\text{lab}}$) have label shift requiring adaptation in direction $\mathbf{v}_{\text{lab}}$. From Proposition 4.4.3, $\mathbf{v}_{\text{feat}}$ primarily involves rates for early/BN layers, while from Proposition 4.4.3, $\mathbf{v}_{\text{lab}}$ involves rates for the last layer. These directions are largely distinct, potentially orthogonal or even opposing in some components if adapting one interferes with the other. The gradient is $\nabla_{\boldsymbol{\alpha}}\mathcal{L}_\mathcal{M}(\boldsymbol{\alpha}) = \frac{1}{2}\nabla_{\boldsymbol{\alpha}}\mathcal{L}_{\mathcal{M}_{\text{feat}}}(\boldsymbol{\alpha}) + \frac{1}{2}\nabla_{\boldsymbol{\alpha}}\mathcal{L}_{\mathcal{M}_{\text{lab}}}(\boldsymbol{\alpha})$. If the optimal strategy for pure feature shift is $\boldsymbol{\alpha}^*_{\text{feat}} \approx c_{\text{feat}}\mathbf{v}_{\text{feat}}$ and for pure label shift is $\boldsymbol{\alpha}^*_{\text{lab}} \approx c_{\text{lab}}\mathbf{v}_{\text{lab}}$, the optimal strategy $\boldsymbol{\alpha}^*_\mathcal{M}$ for the mixed case will be some compromise.

The theorem claims $\boldsymbol{\alpha}^*_\mathcal{S} \cdot \boldsymbol{\alpha}^*_\mathcal{M} < 0$. Let's assume $\mathcal{S}$ is pure feature shift, so $\boldsymbol{\alpha}^*_\mathcal{S} \approx \boldsymbol{\alpha}^*_{\text{feat}}$. The optimal solution $\boldsymbol{\alpha}^*_\mathcal{M}$ minimizes the average loss $\frac{1}{2}(\mathcal{L}_{\mathcal{M}_{\text{feat}}} + \mathcal{L}_{\mathcal{M}_{\text{lab}}})$. If adapting strongly for feature shift (i.e., moving far in direction $\mathbf{v}_{\text{feat}}$) significantly increases the loss for label shift clients, and vice versa, the optimal $\boldsymbol{\alpha}^*_\mathcal{M}$ might involve components that oppose $\boldsymbol{\alpha}^*_{\text{feat}}$. For example, if $\mathbf{v}_{\text{feat}}$ involves increasing rates for early layers and $\mathbf{v}_{\text{lab}}$ involves increasing rates for the last layer, but increasing early layer rates negatively impacts label shift adaptation, $\boldsymbol{\alpha}^*_\mathcal{M}$ might have smaller or even negative components corresponding to early layers compared to $\boldsymbol{\alpha}^*_{\text{feat}}$. If we consider the gradient directions near $\boldsymbol{\alpha} = 0$: $\nabla\mathcal{L}_\mathcal{S} \approx -\mathbf{v}_{\text{feat}}$ and $\nabla\mathcal{L}_\mathcal{M} \approx -\frac{1}{2}(\mathbf{v}_{\text{feat}} + \mathbf{v}_{\text{lab}})$. The optimal points satisfy $\nabla\mathcal{L}(\boldsymbol{\alpha}^*) = 0$. By strong convexity, $\boldsymbol{\alpha}^*$ is roughly proportional to the negative gradient direction at 0. So, $\boldsymbol{\alpha}^*_\mathcal{S} \propto \mathbf{v}_{\text{feat}}$ and $\boldsymbol{\alpha}^*_\mathcal{M} \propto \mathbf{v}_{\text{feat}} + \mathbf{v}_{\text{lab}}$. Their dot product is $\boldsymbol{\alpha}^*_\mathcal{S} \cdot \boldsymbol{\alpha}^*_\mathcal{M} \propto \mathbf{v}_{\text{feat}} \cdot (\mathbf{v}_{\text{feat}} + \mathbf{v}_{\text{lab}}) = \|\mathbf{v}_{\text{feat}}\|^2 + \mathbf{v}_{\text{feat}} \cdot \mathbf{v}_{\text{lab}}$. This dot product is negative if $\mathbf{v}_{\text{feat}} \cdot \mathbf{v}_{\text{lab}} < -\|\mathbf{v}_{\text{feat}}\|^2$, meaning the adaptation directions required by the two shift types are strongly opposing. While not always strictly negative, the optimal strategies are fundamentally different, and applying the strategy optimal for one type can be detrimental for the other, motivating the need for a mechanism like ARN that can learn the appropriate compromise or context-specific strategy. The negative correlation observed empirically (Fig 3) supports this dichotomy.

