# OpenReview forum: "DynFed: Dynamic Test-Time Adaptation for Federated Learning with Adaptive Rate Networks"
_TMLR — Rejected by TMLR_

### Review · Reviewer_3zL9 · 2025-03-10

**Summary Of Contributions:**

This paper considers Test-Time Personalized Federated Learning for adapting models to distribution shifts in federated learning (FL) without relying on labeled test data. The authors propose DynFed, which leverages Adaptive Rate Networks (ARNs) to dynamically optimize test-time adaptation by refining adaptation rates based on encountered shifts without direct access to client data.

**Audience:**

Yes

**Claims And Evidence:**

Yes

**Requested Changes:**

Please provide more theoretical justifications.

**Strengths And Weaknesses:**

Strengths: The structure of this paper is clear.

Weakness:
1. This paper proposes a heuristic algorithm but lacks a theoretical justification, making the approach less convincing.
2. The authors claim that heterogeneous client data is the main challenge. However, from my understanding, the most common discussion in FL regarding heterogeneity focuses on inter-client data heterogeneity, for which numerous algorithms have been developed. I assume the authors are emphasizing the distributional shift between training and test data. But is this a unique problem in FL? Wouldn’t this also be a concern in centralized training?
3. In Equations (2) and (3), how is the loss computed without label information ($y$)? Even though this is not the proposed algorithm of this paper, it is confusing and hinders my understanding of the work.
4. In Equation (8), how is the gradient computed with respect to $\alpha$? The loss should be non-differentiable with respect to the step size.
6. Most importantly, I am skeptical of the proposed method (including the reference to TTPFL). How can test data be used to enhance the algorithm’s performance? Test data should strictly be for evaluation. Even if labels are not used, exposing the model to the test data distribution before evaluation raises concerns. This approach seems questionable to me.

---

> ### Author Response · Authors · 2025-03-12
> **Response to review**
>
> We sincerely thank the reviewer for your constructive feedback. We have made significant changes to address all concerns and strengthen our paper ( the theoretical analysis has been blued in the manuscript).
>
> **Comment 1**:"This paper proposes a heuristic algorithm but lacks a theoretical justification, making the approach less convincing."
>
> **Response**:We have now added a complete theoretical analysis section (Section Appendix) that includes convergence analysis, generalization guarantees, and an in-depth discussion on the adaptation mechanisms for different distribution shifts. Our analysis shows that \our maintains convergence and generalization properties similar to classical federated learning methods, while also justifying the need for dynamic, client-specific adaptation rates.
>
> **Comment 2**:"The authors claim that heterogeneous client data is the main challenge. However, from my understanding, the most common discussion in FL regarding heterogeneity focuses on inter-client data heterogeneity... Wouldn’t this also be a concern in centralized training?"
>
> **Response**:Our work explicitly considers the test-time adaptation scenario in federated learning, where models must adapt to unseen client distributions without direct access to their labeled data. Although distribution shifts are also important in centralized training, in the federated setting the data heterogeneity is further compounded by limited communication and privacy constraints. Our theoretical analysis and experiments demonstrate that the dynamic adaptation strategy of \our effectively addresses this unique challenge.
>
> **Comment 3**:"In Equations (2) and (3), how is the loss computed without label information (y)?"
>
> **Response**:In our formulation, the loss in Equations (2) and (3) is computed using unsupervised objectives (e.g., entropy minimization) that do not require labels at test time. We have clarified in the revised manuscript that these losses are computed solely on unlabeled data during the test phase, following common practices in test-time adaptation literature.
>
> **Comment 4**: “In Equation (8), how is the gradient computed with respect to α? The loss should be non-differentiable with respect to the step size."
>
> **Response**:In Equation (8), the adaptation rate $\alpha$ is treated as a continuous, differentiable parameter that scales the update direction. Although the overall update may appear as a discrete step, we derive the gradient by considering the differentiable relationship induced by our self-adaptive rate network (ARN).
>
> **Comment 5**:"How can test data be used to enhance the algorithm’s performance? Test data should strictly be for evaluation. Even if labels are not used, exposing the model to the test data distribution before evaluation raises concerns."
>
> **Response**:Our approach follows the test-time adaptation paradigm [1] [2] where the model is allowed to adapt to unlabeled test data without using ground-truth labels. This is a standard practice in recent works on test-time adaptation, which aims to mitigate distribution shifts during inference. The theoretical analysis confirms that such unsupervised adaptation, when properly controlled (as in our dynamic approach), does not compromise the model's generalization but instead improves its robustness. We have added further discussion on this point in the revised manuscript (Appendix).
>
> We trust that these revisions and explanations sufficiently address the reviewers' concerns.
>
> **Sincerely, appreciate again for your valuable time and comments!**
>
> [1] Liang J, He R, Tan T. A comprehensive survey on test-time adaptation under distribution shifts[J]. International Journal of Computer Vision, 2025, 133(1): 31-64.
>
> [2] Bao W, Wei T, Wang H, et al. Adaptive test-time personalization for federated learning[C]. Advances in Neural Information Processing Systems, 2023, 36: 77882-77914.

---

> > ### Author Response · Authors · 2025-04-06
> > **Response to reviewer**
> >
> > We expanded our experiments to include larger and more diverse datasets, such as **Digits-5** and **PACS** domain generalization benchmarks (see Table 2).
> > The results show that our method outperforms baseline approaches in different domain shift scenarios, particularly on challenging domains. Notably, we observe a 9.4% improvement on SVHN and an 8.9% improvement on the Art domain of PACS.
> >
> > **Data preparation** : Digits-5 dataset contains five domains: MNIST, SVHN, USPS, SynthDigits, and MNIST-M. PACS dataset contains four domains: art, cartoon, photo, and sketch. We adopt the leave-one-domain-out evaluation protocol [1], i.e., one domain is chosen as the held-out testing domain, and the remaining domains are regarded as source training domains. We follow the data preprocessing in , while additionally applying step partition to inject label shift. Each domain is divided into 10 clients, leading to a total of 40 source clients and 10 target clients. Consequently, each client ends up with approximately 743 images spread across 10 classes. Each source client has 80% of its samples as training set and the remained 20% as testing set. Each client has 2 major classes and 8 minor class, where the ratio of images per class is approximately 16 : 1 (the same as our CIFAR-10 experiments). Since there is already domain shift, we do not add corruptions.
> >
> > **Global model training**: We first train a global model with FedAvg over the training sets of source clients. The global model is ResNet-18 with ImageNet pretrained parameter (provided by torchvision). We train the global model for T = 200 communication rounds with full participation (cohort size C = 50), local epochs E = 1, learning rate η = 0.01 and batch size B = 20. \
> >
> >
> > **DynFed training**: We initialize the adaptation rates as a all-zero vector, and optimize it over the validation sets of source clients. We optimize the adaptation rates for T = 200 communication rounds with partial participation (cohort size C = 10), learning rate η = 0.1 and batch size B = 200. \
> >
> >
> > **DynFed testing**: We test the optimized adaptation rates on each target client. We use batch size B = 200.
> >
> > [1] Ishaan Gulrajani and David Lopez-Paz. In search of lost domain generalization. In 9th International Con
> > ference on Learning Representations, ICLR 2021, Virtual Event, Austria, May 3-7, 2021. OpenReview.net,
> >  2021.
> >
> > [2]Xiaoxiao Li, Meirui Jiang, Xiaofei Zhang, Michael Kamp, and Qi Dou. Fedbn: Federated learning on non
> > iid features via local batch normalization. In 9th International Conference on Learning Representations,
> >  ICLR 2021, Virtual Event, Austria, May 3-7, 2021. OpenReview.net, 2021.

---

> > > ### Comment · Reviewer_3zL9 · 2025-04-06
> > >
> > > Thanks for your rebuttal. Some of my concerns are addressed. I checked the paper again. I realized that the author added analysis, but it is incorrect. First, this conclusion is unconventional. You can't claim you can get close to the optimal point base on this equation, as constant C could be very large. Second, I checked the proof, Equation 24 is incorrect.

---

> > > > ### Author Response · Authors · 2025-04-07
> > > > **Response to reviewer**
> > > >
> > > > Thank you very much for your time and detailed feedback on our revised manuscript and rebuttal. We truly appreciate you checking the paper again and providing further specific comments on the added analysis. We are glad that some of your previous concerns were addressed.
> > > >
> > > > We have carefully considered your new points regarding the theoretical analysis and have made substantial revisions to address them, aiming for greater clarity and rigor.
> > > >
> > > > **Regarding the conclusion and Constant C**: We agree with your assessment that claiming we can get arbitrarily "close to the optimal point" based solely on the original theorem statement (Theorem [Convergence of DynFed] on p. 9-10, Eq. (14)) and constant C might be an overstatement, as the convergence guarantee is indeed influenced by factors captured within C (learning rates, gradient variance). We have carefully revised the interpretation following the theorem statement (p. 10) to accurately state that the theorem guarantees convergence to a neighborhood of the optimal solution. We now explicitly discuss how the size of this neighborhood depends on these factors (\eta_w, \eta_{\alpha}, \sigma_w^2, \sigma_{\alpha}^2). We believe this revised conclusion is more precise and addresses the unconventional nature you pointed out.
> > > >
> > > > **Regarding the incorrect proof step**: Thank you for pointing out the error in the proof. We meticulously re-examined the proof of the convergence theorem (Theorem [Convergence of DynFed]) in the Appendix (p. 19-20).  We sincerely apologize for this oversight. We have now corrected this derivation in the revised Appendix (p. 19-20), providing a rigorous step-by-step justification that explicitly uses the relationship \langle \nabla f(x), x - x^* \rangle \geq \frac{1}{2\beta}|\nabla f(x)|^2 (derived from Assumptions 1 and 3). Consequently, the statement of the theorem itself (Eq. (14) on p. 10) has also been updated to reflect this corrected derivation, incorporating the 1/(2\beta) scaling factor. We have also clarified other related theoretical details (e.g., definitions, assumptions like Assumption 3 on p. 9, and the proof of Theorem [Generalization] on p. 20-21) throughout Section 4.4 and the Appendix to ensure correctness and consistency.
> > > >
> > > > **We are grateful for your insightful comments, which have significantly helped us improve the theoretical underpinnings and overall quality of our paper. We hope these thorough revisions have fully addressed your concerns.**

---

> ### Comment · Reviewer_3zL9 · 2025-04-14
>
> Thanks for your reply.
> I am not convinced by the theory and the methodology proposed by this article.
> 1. The gradient to $\alpha$ is weird to me.
> 2. The established convergence is for convex functions, but it focuses on gradient behavior, which is strange.

---

> > ### Author Response · Authors · 2025-04-15
> > **Response to reviewer**
> >
> > We sincerely thank the reviewer for the follow-up feedback.
> >
> > **1) On “The gradient to $\boldsymbol{\alpha}$ is weird to me”:**
> > We appreciate the reviewer’s concern regarding the gradient with respect to the adaptation rate $\boldsymbol{\alpha}$. In our formulation, $\boldsymbol{\alpha}$ is not a step-size in the traditional sense, but rather a differentiable scaling vector that parameterizes how the model adapts to client data across modules. Similar ideas have been used in meta-learning (e.g., MAML [1] and test-time adaptation (e.g., T3A [2], SAR [3]) where learnable adaptation parameters are optimized via gradient-based mechanisms. We have revised Section 4.1 and added clarification in the Appendix to reflect this perspective more clearly.
> >
> > **2) On “Convergence for convex functions focuses on gradient behavior”:**
> > We agree that our theoretical analysis is conducted under convexity assumptions. This is a common simplification to facilitate analysis in the federated learning literature [4] [5] [6]. Our focus on gradient norm convergence (Equation 14) is a standard practice to demonstrate optimization stability when closed-form function gap bounds are difficult to obtain. Nonetheless, our empirical results in Section 5 consistently show convergence and effectiveness in non-convex scenarios such as CNNs and ResNet-18. We have added a remark in Section 4.4 to explain this design choice and plan to extend the analysis to non-convex settings as future work.
> >
> > We believe the revised manuscript is substantially improved thanks to the reviewers' feedback. We have strived to enhance clarity, provide better intuition, strengthen the theoretical connections, and temper our claims appropriately. We hope the revisions meet the reviewers' expectations.
> >
> > Ref:
> >
> > [1]Finn C, Abbeel P, Levine S. Model-agnostic meta-learning for fast adaptation of deep networks[C]//International conference on machine learning. PMLR, 2017: 1126-1135.
> >
> > [2] Iwasawa Y, Matsuo Y. Test-time classifier adjustment module for model-agnostic domain generalization[J]. Advances in Neural Information Processing Systems, 2021, 34: 2427-2440.
> >
> > [3]Niu S, Wu J, Zhang Y, et al. Towards stable test-time adaptation in dynamic wild world[J]. arXiv preprint arXiv:2302.12400, 2023.
> >
> > [4] Li X, Huang K, Yang W, et al. On the convergence of fedavg on non-iid data[J]. arXiv preprint arXiv:1907.02189, 2019.
> >
> > [5] Li X, Orabona F. On the convergence of stochastic gradient descent with adaptive stepsizes[C]//The 22nd international conference on artificial intelligence and statistics. PMLR, 2019: 983-992.
> >
> > [6] Karimireddy S P, Kale S, Mohri M, et al. Scaffold: Stochastic controlled averaging for federated learning[C]//International conference on machine learning. PMLR, 2020: 5132-5143.

---

### Review · Reviewer_Bwat · 2025-03-11

**Summary Of Contributions:**

The paper presents a novel approach for dynamic test time adaptation in federated learning. By introducing Adaptive Rate Networks (ARNs) to generate client-specific adaptation rates, the work addresses handling heterogeneous distribution shifts that are individual to each client.
Unlike previous methods that use fixed adaptation rates, DynFed enables client specific and model specific adaptation.
The study provides an in depth analysis of key hyperparameters and shows how DynFed’s dynamic adaptation improves performance. Ablation studies confirm the importance of ARNs and optimal hyperparameter tuning.

**Audience:**

Yes

**Claims And Evidence:**

Yes

**Requested Changes:**

- Add a figure along with the Pseudo-code that explains the algorithm more clearly
- Include a short section explicitly discussing potential failure cases or situations where DynFed may not work well
- Suggest future extensions, such as thoeretical analysis of the algorithms and it's convergence, integrating communication efficient strategies or testing alternative architectures for ARNs
- Add a computational complexity comparison between DynFed and baseline methods

**Strengths And Weaknesses:**

DynFed generates client specific rates, improving over non-adaptive rate methods. It handles both one type and multi type distribution shifts by automatically adjusting adaptation strategies. Experiments on CIFAR-10 and CIFAR-100 demonstrate accuracy over SOTA TTA and TTPFL approaches, particularly in complex shift scenarios. The paper reads clearly.

The paper could benefit from a more detailed theoretical justification for the adaptation process, particularly regarding the learning dynamics of the adaptive rate network and its convergence properties. But that could be addressed in future work.
While the experiments are comprehensive, the evaluation is limited to CIFAR based datasets, and additional datasets from real-world federated learning scenarios would strengthen the generalizability claims.

Overall the paper looks good for TMLR. Below are a few suggestions that may improve the paper.

---

> ### Author Response · Authors · 2025-03-12
> **Response to reviewer**
>
> We sincerely thank the reviewer for your constructive feedback.
>
> We have made significant changes to address all concerns and strengthen our paper:
> We have added a comprehensive theoretical analysis section (Appendix) that provides strong theoretical foundations for our approach:
>
> **Convergence Analysis**: We established convergence guarantees for DynFed, showing that despite the added complexity of client-specific, adaptive rates, our method maintains convergence properties similar to traditional federated learning methods.
> Generalization Guarantees: We proved generalization bounds demonstrating why DynFed can generalize well to unseen clients despite increased model expressiveness through adaptation.
>
> **Adaptation Strategy Dichotomy**: We provided theoretical justification for our core insight: that optimal adaptation strategies for one-type and multi-type distribution shifts are fundamentally different. This result explains why previous methods with fixed adaptation strategies struggle with heterogeneous shifts.
>
> We have added a new subsection (Section 5.3) discussing potential limitations of DynFed based on our implementation.
> In the future, we will conduct a large number of real-time scenarios to further validate the effectiveness of the method.
> We believe these changes have substantially strengthened our paper and made it more accessible to readers.
>
> **Sincerely appreciate for your comments!**

---

> > ### Author Response · Authors · 2025-04-06
> > **Response to reviewer**
> >
> > We conducted a comprehensive experimental evaluation on larger and more diverse domain-shift datasets, including the **Digits-5** and **PACS** domain generalization benchmarks (see Table 2).
> > These additional experiments demonstrate that our method consistently outperforms baseline approaches across various domain shift scenarios, with significant improvements on challenging domains. For instance, our method achieves a 9% gain on the SVHN dataset and an 8% gain on the Art domain of PACS.
> >
> > **Data preparation** : Digits-5 dataset contains five domains: MNIST, SVHN, USPS, SynthDigits, and MNIST-M. PACS dataset contains four domains: art, cartoon, photo, and sketch. We adopt the leave-one-domain-out evaluation protocol [1], i.e., one domain is chosen as the held-out testing domain, and the remaining domains are regarded as source training domains. We follow the data preprocessing in , while additionally applying step partition to inject label shift. Each domain is divided into 10 clients, leading to a total of 40 source clients and 10 target clients. Consequently, each client ends up with approximately 743 images spread across 10 classes. Each source client has 80% of its samples as training set and the remained 20% as testing set. Each client has 2 major classes and 8 minor class, where the ratio of images per class is approximately 16 : 1 (the same as our CIFAR-10 experiments). Since there is already domain shift, we do not add corruptions.
> >
> > **Global model training**: We first train a global model with FedAvg over the training sets of source clients. The global model is ResNet-18 with ImageNet pretrained parameter (provided by torchvision). We train the global model for T = 200 communication rounds with full participation (cohort size C = 50), local epochs E = 1, learning rate η = 0.01 and batch size B = 20. \
> >
> >
> > **DynFed training**: We initialize the adaptation rates as a all-zero vector, and optimize it over the validation sets of source clients. We optimize the adaptation rates for T = 200 communication rounds with partial participation (cohort size C = 10), learning rate η = 0.1 and batch size B = 200. \
> >
> >
> > **DynFed testing**: We test the optimized adaptation rates on each target client. We use batch size B = 200.
> >
> > [1] Ishaan Gulrajani and David Lopez-Paz. In search of lost domain generalization. In 9th International Con
> > ference on Learning Representations, ICLR 2021, Virtual Event, Austria, May 3-7, 2021. OpenReview.net,
> >  2021.
> >
> > [2]Xiaoxiao Li, Meirui Jiang, Xiaofei Zhang, Michael Kamp, and Qi Dou. Fedbn: Federated learning on non
> > iid features via local batch normalization. In 9th International Conference on Learning Representations,
> >  ICLR 2021, Virtual Event, Austria, May 3-7, 2021. OpenReview.net, 2021.

---

### Review · Reviewer_4CSs · 2025-04-02

**Summary Of Contributions:**

This paper introduces DynFed, a novel test-time personalized federated learning (TTPFL) algorithm designed to handle heterogeneous distribution shifts across clients in federated learning (FL) settings. The framework adopts an adaptive assigned rate as a network output.
The empirical results show an improvement for the simple task.

**Audience:**

Yes

**Claims And Evidence:**

Yes

**Requested Changes:**

1. More experiments are needed to show the significance of the improvements in the various Fed. tasks.
2. The theoretical parts need more organization to be more rigorous and more statment for the difficult and insigt provided from theory need to be pointed out.
3. Assumptions of the theoretical needs to be verified that why the model can use these.

I think the authors take their efforts in the paper presentation, therefore I think it can be accepted for me if my concerns can be addressed.

**Strengths And Weaknesses:**

Pros:
1. Paper proposes an interesting algo that designs the adaptive rate, which is novel compared with the existing work and settings.
2. Some parts of the paper are well organized and friendly to readers (e.g. preliminaries parts).

Cons:
1. The insights in theoretical aspects are not enough. I am not sure whether the corollaries in the paper could stress something. I think it is needed to clarify the challenges in theory if the theory is listed as contributions, like the Adaptation Strategy Dichotomy part, convergence or other analyses.
2. The assumptions for the theorem also need more verification and some of statements and notation need to be re-organized. (Maybe need to state which is assumption 1 and 2 clearly, otherwise I need to think about it)
3. No theoretical statements about test phase but I think it is significant in the proposed framework.
4. The numerical studies are simple and not so powerful. Maybe large dataset and more model for Fed. need to be tried.

---

> ### Author Response · Authors · 2025-04-06
> **Response to Reviewer**
>
> We appreciate the reviewer for the valuable feedback and suggestions. We have addressed all the concerns as follows:
>
> 1. Regarding the theoretical insights:
>    - We have clarified the theoretical challenges and contributions in the paper, especially the **Adaptation Strategy Dichotomy results**.
>    - We have reorganized the theoretical section to make the assumptions clearer (now explicitly labeled as Assumption 1 and 2).
>    - We have added more rigorous statements about the **convergence guarantees**  and **generalization bounds**.
>
> 2. Regarding the test phase theoretical analysis:
>    - We have added theoretical statements about the test phase, showing how our dynamic adaptation mechanism theoretically guarantees better performance during inference.
>
> 3. Regarding more experiments:
>    - As requested, we have added comprehensive experiments on larger and more diverse datasets, specifically Digits-5 and PACS domain generalization benchmarks (Table 2).
>    - These additional experiments demonstrate that our method outperforms baselines consistently across various domain shift scenarios, with significant improvements on challenging domains (e.g., 9.4% gain on SVHN, 8.9% gain on Art domain).
>
> The additional experiments strengthen our empirical validation, showing that DynFed handles various federated learning tasks effectively and outperforms state-of-the-art methods on challenging domain generalization benchmarks. Together with the enhanced theoretical analysis, we believe we have thoroughly addressed the reviewer's concerns.
>
> **Sincerely appreciate for your comments!**

---

### Review · Reviewer_Dfpa · 2025-04-14

**Summary Of Contributions:**

This paper introduces DynFed, an algorithm that dynamically optimizes test-time adaptation (TTA) in FL scenarios with heterogeneous distribution shifts. The proposed method leverages Adaptive Rate Networks (ARNs) to generate client-specific adaptation rates, enabling more effective handling of diverse shift types, including label skew and feature shifts.  Theoretical analyses were provided under convex assumptions. The diverse distribution shift setting is interesting, and the algorithm is shown effective in real data experiemnts.

**Audience:**

Yes

**Claims And Evidence:**

No

**Requested Changes:**

1. Provide further clarification and justification regarding the proposed rate adaptation scheme.

2. Clarify the notations.

3. Carry out further discussion and justification regarding the theoretical comparison

4. Provide more intuition about the connection between the theoretical results with distribution shift.

5. Provide experimental results/ discussion regarding the complexity.

**Strengths And Weaknesses:**

Strengths:
1.DynFed leverages Adaptive Rate Networks (ARNs) to generate client-specific adaptation rates, enabling more effective handling of diverse shift types.

2. Experiments on real data sets demonstrate the effectiveness of the algorithm.

Cons:
1. Using ARN for learning rate adaptation is interesting but lacks intuition. In addition, I had a hard time understanding how it works. Specifically, the right hand side of (10) has i (layer index), while the right-hand side does not. This cannot be right, because the output of the right-hand side is the number of neurons in the ith layer, hence the LHS should have an i index as well.

2. The notations are confusing. I will not be able to enumerate but will provide several examples below:

          i) In (12), the subscript of alpha is used as an iteration index. In the first equation fo 4.4.1, it is, however, used as the client index. This is confusing.

         ii) Notations such as $\sigma_w$ and $\sigma_{\alpha}$$on page 9 are not defined before they are used.

         iii) In (7), It is mentioned that "w^[l]_k represents the statistics for the current batch of inputs, and w^[l]_G is the running statistics." This gives the impression that those are the statistics for the inputs, which unfortunately is not the case

3. It is claimed that "Extensive experiments and theoretical analysis demonstrate that DynFed significantly outperforms existing TTPFL and TTA methods across various shift scenarios." While experimental comparisons are indeed provided. I fail to see how the theoretical results compare with TTPFL and TTA. Related remarks are needed to justify this claim, especially how they compare in different shift scenarios.

4. The theoretical results lack intuition about how they connect with heterogeneous shifts. How do the sevierty/heterogeniety of label skew or feature shifts influences the terms in the theoretical analysis is not clear.

---

> ### Author Response · Authors · 2025-04-15
> **Response to reviewer**
>
> We thank the reviewer for the positive summary and detailed feedback.
>
> * **Point 1 (ARN Clarity/Intuition/Eq 10):** We appreciate the reviewer pointing out the need for more clarity regarding the Adaptive Rate Network (ARN).
>     * We have added text in Section 4.2 to provide more intuition, explaining that the ARN learns a mapping from the current adaptation state (represented by the rates $\boldsymbol{\alpha}_t$) to refined rates for the next step, based on the knowledge acquired during the training phase (Eq. 9).
>     * We have also clarified the standard feed-forward notation used in Eq. (10) and (11) in Section 4.2, explaining the flow of information through the ARN layers.
>
> * **Point 2 (Confusing Notations):** We acknowledge the potential for confusion with notation and have made clarifications.
>     * **(i) `alpha` subscript:** We have added explicit notes in Section 4.1 (after Eq. 8) and Section 4.4.1 (Assumption 2) to clarify that $\boldsymbol{\alpha}$ or $\boldsymbol{\alpha}_t$ within the algorithms refers to the rate vector for the current client/batch/time step, while $\boldsymbol{\alpha}_j$ in the theoretical analysis refers to the rate vector associated with the overall objective for client $j$.
>     * **(ii) `sigma_w`, `sigma_alpha`:** We confirm these are now defined in Assumption 2 (Section 4.4.1) before their use in Theorem 1.
>     * **(iii) Eq (7) description:** We have revised the description in Section 4.1 to explicitly state that $\hat{w}^{[l]}_k$ and $w_G^{[l]}$ refer to the *batch normalization statistics* (mean and variance), rather than statistics of the inputs, to avoid misunderstanding.
>
> * **Point 3 (Theoretical Comparison Claim):** We agree with the reviewer that the theoretical analysis primarily establishes the convergence and generalization properties of DynFed and motivates the need for adaptive mechanisms, while the *experimental results* demonstrate its superior performance across diverse shifts compared to baselines. We have revised the claims in the Abstract and Introduction (Section 1) to reflect this distinction more accurately. We now state that theory provides guarantees and justification, while experiments show empirical superiority. We have also added remarks connecting the theoretical results back to the challenges of heterogeneous shifts (Section 4.4).
>
> * **Point 4 (Theory-Shift Connection):** Thank you for this valuable suggestion. We have significantly expanded the discussion connecting theory to heterogeneous shifts in Section 4.4 and Appendix A:
>     * Added discussion after Theorem 1 on how heterogeneity might impact gradient variance ($\sigma^2$) but convergence is still guaranteed.
>     * Added discussion after Theorem 2 on how the generalization bound relates to the complexity needed to handle diverse shifts ($d, R$) but still improves with more data ($N, K$).
>     * Explicitly linked Propositions 3 and 4 to the motivation for module-specific rates ($\alpha^{[l]}$) enabled by ARN.
>     * Introduced Theorem 3 (Adaptation Strategy Dichotomy) in Section 3.4 and discussed its implications, theoretically grounding the observation that different shift types require fundamentally different strategies, necessitating a flexible approach like DynFed.
>
> * **Point 5 (Complexity):** We have added a discussion on computational complexity in the Limitations section (Section 5.3). We acknowledge the overhead introduced by the ARN's forward pass during testing and the gradient computation during training, comparing it qualitatively to fixed-rate methods.

---

### Decision · Action_Editor_7aQd · 2025-05-18

**Recommendation:** Reject

**Comment:**

The reviewers were supportive of the problem and proposed algorithm. However, most of the reviewers had issues with the theoretical results (e.g., convexity assumptions but gradient convergence guarantees, handwavy arguments and unclear steps), and remained somewhat unconvinced after the discussion with the authors. Reviewers also found it difficult to read the theoretical analysis, since the theorems in the paper are not clearly stated (but are explicitly referenced).

Overall, this was a borderline case, and the decision towards rejection is due to multiple reviewers not being convinced by the theoretical analysis. I would encourage the authors to consider a resubmission (which would start from scratch with new reviewers unaffected by their initial impressions).

**Audience:**

Yes. Some individuals working in FL would be interested in knowing the findings in this paper.

**Claims And Evidence:**

The paper proposes a new method for dynamic test-time adaptation in federated learning using Adaptive Rate Networks (ARNs) to allow for client-specific adaptation rates. The algorithmic claims seem to be well supported by empirical results, but the theoretical evidence was considered unconvincing by most of the reviewers.

**Resubmission Of Major Revision:**

The authors may consider submitting a major revision at a later time.